# Tensor Biclustering

**Soheil Feizi**
Stanford University
sfeizi@stanford.edu

**Hamid Javadi**
Stanford University
hrhakim@stanford.edu

**David Tse**
Stanford University
dntse@stanford.edu

## Abstract

Consider a dataset where data is collected on multiple features of multiple individuals over multiple times. This type of data can be represented as a three dimensional individual/feature/time tensor and has become increasingly prominent in various areas of science. The tensor biclustering problem computes a subset of individuals and a subset of features whose signal trajectories over time lie in a low-dimensional subspace, modeling similarity among the signal trajectories while allowing different scalings across different individuals or different features. We study the information-theoretic limit of this problem under a generative model. Moreover, we propose an efficient spectral algorithm to solve the tensor biclustering problem and analyze its achievability bound in an asymptotic regime. Finally, we show the efficiency of our proposed method in several synthetic and real datasets.

## 1 Introduction

Let $\mathbf{T} \in \mathbb{R}^{n_1 \times n_2}$ be a data matrix whose rows and columns represent individuals and features, respectively. Given $\mathbf{T}$, the matrix biclustering problem aims to find a subset of individuals (i.e., $J_1 \subset \{1, 2, ..., n_1\}$) which exhibit similar values across a subset of features (i.e., $J_2 \subset \{1, 2, ..., n_2\}$) (Figure 1-a). The matrix biclustering problem has been studied extensively in machine learning and statistics and is closely related to problems of sub-matrix localization, planted clique and community detection [1, 2, 3].

In modern datasets, however, instead of collecting data on every individual-feature pair at a single time, we may collect data at multiple times. One can visualize a *trajectory* over time for each individual-feature pair. This type of datasets has become increasingly prominent in different areas of science. For example, the roadmap epigenomics dataset [4] provides multiple histon modification marks for genome-tissue pairs, the genotype-tissue expression dataset [5] provides expression data on multiple genes for individual-tissue pairs, while there have been recent efforts to collect various omics data in individuals at different times [6].

Suppose we have $n_1$ individuals, $n_2$ features, and we collect data for every individual-feature pair at $m$ different times. This data can be represented as a three dimensional tensor $\mathcal{T} \in \mathbb{R}^{n_1 \times n_2 \times m}$ (Figure 1-b). The *tensor biclustering* problem aims to compute a subset of individuals and a subset of features whose trajectories are highly similar. Similarity is modeled as the trajectories as lying in a low-dimensional (say one-dimensional) subspace (Figure 1-d). This definition allows different scalings across different individuals or different features, and is important in many applications such as in omics datasets [6] because individual-feature trajectories often have their own intrinsic scalings. In particular, at each time the individual-feature data matrix may not exhibit a matrix bicluster separately. This means that repeated applications of matrix biclustering cannot solve the tensor biclustering problem. Moreover, owing to the same reason, trajectories in a bicluster can have large distances among themselves (Figure 1-d). Thus, a distance-based clustering of signal trajectories is likely to fail as well.

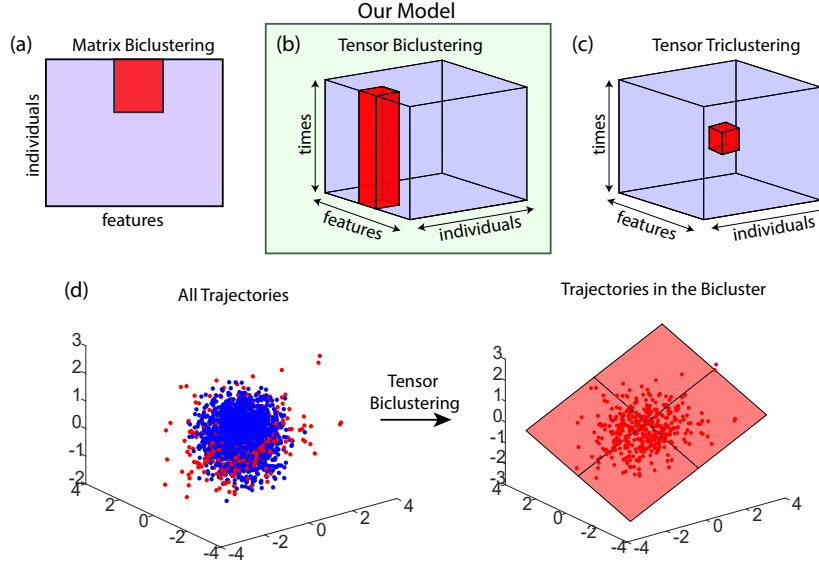

Figure 1: (a) The matrix biclustering problem. (b) The tensor biclustering problem. (c) The tensor triclustering problem. (d) A visualization of a bicluster in a three dimensional tensor. Trajectories in the bicluster (red points) form a low dimensional subspace.

This problem formulation has two main differences with tensor *triclustering*, which is a natural generalization of matrix biclustering to a three dimensional tensor (Figure 1-c). Firstly, unlike tensor triclustering, tensor biclustering has an asymmetric structure along tensor dimensions inspired by aforementioned applications. That is, since a tensor bicluster is defined as a subset of individuals and a subset of features with similar trajectories, the third dimension of the tensor (i.e., the time dimension) plays a different role compared to the other two dimensions. This is in contrast with tensor triclustering where there is not such a difference between roles of tensor dimensions in defining the cluster. Secondly, in tensor biclustering, the notion of a cluster is defined regarding to trajectories lying in a low-dimensional subspace while in tensor triclustering, a cluster is defined as a sub-cube with similar entries.

Finding statistically significant patterns in multi-dimensional data tensors has been studied in dimensionality reduction [7, 8, 9, 10, 11, 12, 13, 14], topic modeling [15, 16, 17], among others. One related model is the spiked tensor model [7]. Unlike the tensor biclustering model that is asymmetric along tensor dimensions, the spiked tensor model has a symmetric structure. Computational and statistical limits for the spiked tensor model have been studied in [8, 9, 10, 14], among others. For more details, see Supplementary Materials (SM) Section 1.3.

In this paper, we study information-theoretic and computational limits for the tensor biclustering problem under a statistical model described in Section 2. From a computational perspective, we present four polynomial time methods and analyze their asymptotic achievability bounds. In particular, one of our proposed methods, namely tensor folding+spectral, outperforms other methods both theoretically (under realistic model parameters) and numerically in several synthetic and real data experiments. Moreover, we characterize a fundamental limit under which no algorithm can solve the tensor biclustering problem reliably in a minimax sense. We show that above this limit, a maximum likelihood estimator (MLE) which has an exponential computational complexity can solve this problem with vanishing error probability.

## 1.1 Notation

We use $\mathcal{T}$, $\mathcal{X}$, and $\mathcal{Z}$ to represent input, signal, and noise tensors, respectively. For any set $J$, $|J|$ denotes its cardinality. $[n]$ represents the set $\{1, 2, ..., n\}$. $\bar{J} = [n] - J$. $\|\mathbf{x}\|_2 = (\mathbf{x}^t\mathbf{x})^{1/2}$ is the second norm of the vector $\mathbf{x}$. $\mathbf{x} \otimes \mathbf{y}$ is the Kronecker product of two vectors $\mathbf{x}$ and $\mathbf{y}$. The asymptotic notation $a(n) = \mathcal{O}(b(n))$ means that, there exists a universal constant $c$ such that for sufficiently

large $n$, we have $|a(n)| < cb(n)$. If there exists $c > 0$ such that $a(n) = \mathcal{O}(b(n)\log(n)^c)$, we use the notation $a(n) = \tilde{\mathcal{O}}(b(n))$. The asymptotic notation $a(n) = \Omega(b(n))$ and $a(n) = \tilde{\Omega}(b(n))$ is the same as $b(n) = \mathcal{O}(a(n))$ and $b(n) = \tilde{\mathcal{O}}(a(n))$, respectively. Moreover, we write $a(n) = \Theta(b(n))$ iff $a(n) = \Omega(b(n))$ and $b(n) = \Omega(a(n))$. Similarly, we write $a(n) = \tilde{\Theta}(b(n))$ iff $a(n) = \tilde{\Omega}(b(n))$ and $b(n) = \tilde{\Omega}(a(n))$.

## 2  Problem Formulation

Let $\mathcal{T} = \mathcal{X} + \mathcal{Z}$ where $\mathcal{X}$ is the signal tensor and $\mathcal{Z}$ is the noise tensor. Consider

$$\mathcal{T} = \mathcal{X} + \mathcal{Z} = \sum_{r=1}^{q} \sigma_r \mathbf{u}_r^{(J_1)} \otimes \mathbf{w}_r^{(J_2)} \otimes \mathbf{v}_r + \mathcal{Z}, \tag{1}$$

where $\mathbf{u}_r^{(J_1)}$ and $\mathbf{w}_r^{(J_2)}$ have zero entries outside of $J_1$ and $J_2$ index sets, respectively. We assume $\sigma_1 \geq \sigma_2 \geq ... \geq \sigma_q > 0$. Under this model, trajectories $\mathcal{X}(J_1, J_2, :)$ form an at most $q$ dimensional subspace. We assume $q \ll \min(m, |J_1| \times |J_2|)$.

**Definition 1** (Tensor Biclustering). *The problem of tensor biclustering aims to compute bicluster index sets $(J_1, J_2)$ given $\mathcal{T}$ according to (1).*

In this paper, we make the following simplifying assumptions: we assume $q = 1$, $n = |n_1| = |n_2|$, and $k = |J_1| = |J_2|$. To simplify notation, we drop superscripts $(J_1)$ and $(J_2)$ from $\mathbf{u}_1^{(J_1)}$ and $\mathbf{w}_1^{(J_2)}$, respectively. Without loss of generality, we normalize signal vectors such that $\|\mathbf{u}_1\| = \|\mathbf{w}_1\| = \|\mathbf{v}_1\| = 1$. Moreover, we assume that for every $(j_1, j_2) \in J_1 \times J_2$, $\Delta \leq \mathbf{u}_1(j_1) \leq c\Delta$ and $\Delta \leq \mathbf{w}_1(j_2) \leq c\Delta$, where $c$ is a constant. Under these assumptions, a signal trajectory can be written as $\mathcal{X}(j_1, j_2, :) = \mathbf{u}_1(j_1)\mathbf{w}_1(j_2)\mathbf{v}_1$. The scaling of this trajectory depends on row and column specific parameters $\mathbf{u}_1(j_1)$ and $\mathbf{w}_1(j_2)$. Note that our analysis can be extended naturally to a more general setup of having multiple embedded biclusters with $q > 1$. We discuss this in Section 7.

Next we describe the noise model. If $(j_1, j_2) \notin J_1 \times J_2$, we assume that entries of the noise trajectory $\mathcal{Z}(j_1, j_2, :)$ are i.i.d. and each entry has a standard normal distribution. If $(j_1, j_2) \in J_1 \times J_2$, we assume that entries of $\mathcal{Z}(j_1, j_2, :)$ are i.i.d. and each entry has a Gaussian distribution with zero mean and $\sigma_z^2$ variance. We analyze the tensor biclustering problem under two noise models for $\sigma_z^2$:

- **Noise Model I:** In this model, we assume $\sigma_z^2 = 1$, i.e., the variance of the noise within and outside of the bicluster is assumed to be the same. This is the noise model often considered in analysis of sub-matrix localization [2, 3] and tensor PCA [7, 8, 9, 10, 11, 12, 14]. Although this model simplifies the analysis, it has the following drawback: under this noise model, for every value of $\sigma_1$, the average trajectory lengths in the bicluster is larger than the average trajectory lengths outside of the bicluster. See SM Section 1.2 for more details.

- **Noise Model II:** In this model, we assume $\sigma_z^2 = \max(0, 1 - \frac{\sigma_1^2}{mk^2})$, i.e., $\sigma_z^2$ is modeled to minimize the difference between the average trajectory lengths within and outside of the bicluster. If $\sigma_1^2 < mk^2$, noise is added to make the average trajectory lengths within and outside of the bicluster comparable. See SM Section 1.2 for more details.

## 3  Computational Limits of the Tensor Biclustering Problem

### 3.1  Tensor Folding+Spectral

Recall the formulation of the tensor biclustering problem (1). Let

$$\mathbf{T}_{(j_1,1)} \triangleq \mathcal{T}(j_1, :, :) \quad \text{and} \quad \mathbf{T}_{(j_2,2)} \triangleq \mathcal{T}(:, j_2, :), \tag{2}$$

be horizontal (the first mode) and lateral (the second mode) matrix slices of the tensor $\mathcal{T}$, respectively. One way to learn the embedded bicluster in the tensor is to compute row and column indices whose trajectories are highly correlated with each other. To do that, we compute

$$\mathbf{C}_1 \triangleq \sum_{j_2=1}^{n} \mathbf{T}_{(j_2,2)}^t \mathbf{T}_{(j_2,2)} \quad \text{and} \quad \mathbf{C}_2 \triangleq \sum_{j_1=1}^{n} \mathbf{T}_{(j_1,1)}^t \mathbf{T}_{(j_1,1)}. \tag{3}$$

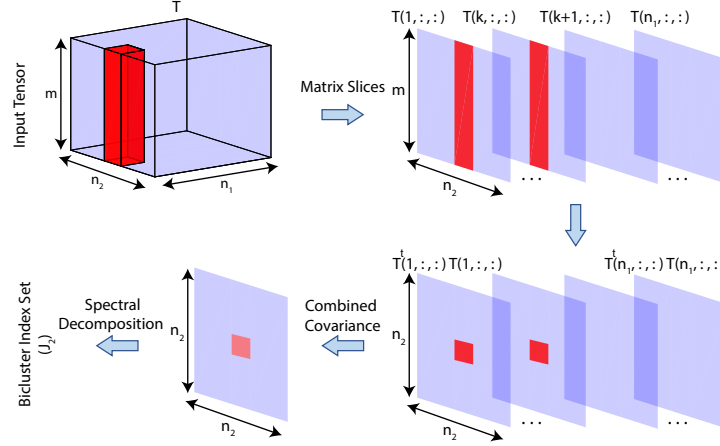

Figure 2: A visualization of the tensor folding+spectral algorithm 1 to compute the bicluster index set $J_2$. The bicluster index set $J_1$ can be computed similarly.

---

**Algorithm 1** Tensor Folding+Spectral

---

**Input:** $\mathcal{T}, k$
Compute $\hat{\mathbf{u}}_1$, the top eigenvector of $\mathbf{C}_1$
Compute $\hat{\mathbf{w}}_1$, the top eigenvector of $\mathbf{C}_2$
Compute $\hat{J}_1$, indices of the $k$ largest values of $|\hat{\mathbf{w}}_1|$
Compute $\hat{J}_2$, indices of the $k$ largest values of $|\hat{\mathbf{u}}_1|$
**Output:** $\hat{J}_1$ and $\hat{J}_2$

---

$\mathbf{C}_1$ represents a combined covariance matrix along the tensor columns (Figure 2). We refer to it as the folded tensor over the columns. If there was no noise, this matrix would be equal to $\sigma_1^2 \mathbf{u}_1 \mathbf{u}_1^t$. Thus, its eigenvector corresponding to the largest eigenvalue would be equal to $\mathbf{u}_1$. On the other hand, we have $\mathbf{u}_1(j_1) = 0$ if $j_1 \notin J_1$ and $|\mathbf{u}_1(j_1)| > \Delta$, otherwise. Therefore, selecting $k$ indices of the top eigenvector with largest magnitudes would recover the index set $J_1$. However, with added noise, the top eigenvector of the folded tensor would be a perturbed version of $\mathbf{u}_1$. Nevertheless one can estimate $J_1$ similarly (Algorithm 1). A similar argument holds for $\mathbf{C}_2$.

**Theorem 1.** *Let $\hat{\mathbf{u}}_1$ and $\hat{\mathbf{w}}_1$ be top eigenvectors of $\mathbf{C}_1$ and $\mathbf{C}_2$, respectively. Under both noise models I and II,*

- *for $m < \tilde{\mathcal{O}}(\sqrt{n})$, if $\sigma_1^2 = \tilde{\Omega}(n)$,*

- *for $m = \tilde{\Omega}(\sqrt{n})$, if $\sigma_1^2 = \tilde{\Omega}(\sqrt{n}\max(n,m))$,*

*as $n \to \infty$, with high probability, we have $|\hat{\mathbf{u}}_1(j_1)| > |\hat{\mathbf{u}}_1(j_1')|$ and $|\hat{\mathbf{w}}_1(j_2)| > |\hat{\mathbf{w}}_1(j_2')|$ for every $j_1 \in J_1$, $j_1' \in \bar{J}_1$, $j_2 \in J_2$ and $j_2' \in \bar{J}_2$.*

In the proof of Theorem 1, following the result of [18] for a Wigner noise matrix, we have proved an $l_\infty$ version of the Davis-Kahan Lemma for a Wishart noise matrix. This lemma can be of independent interest for the readers.

## 3.2 Tensor Unfolding+Spectral

Let $\mathbf{T}_{unfolded} \in \mathbb{R}^{m \times n^2}$ be the unfolded tensor $\mathcal{T}$ such that $\mathbf{T}_{unfolded}(:, (j_1 - 1)n + j_2) = \mathcal{T}(j_1, j_2, :)$ for $1 \leq j_1, j_2 \leq n$. Without noise, the right singular vector of this matrix is $\mathbf{u}_1 \otimes \mathbf{w}_1$ which corresponds to the singular value $\sigma_1$. Therefore, selecting $k^2$ indices of this singular vector with largest magnitudes would recover the index set $J_1 \times J_2$. With added noise, however, the top singular vector of the unfolded tensor will be perturbed. Nevertheless one can estimate $J_1 \times J_2$ similarly (SM Section 2).

**Theorem 2.** *Let $\hat{\mathbf{x}}$ be the top right singular vector of $\mathbf{T}_{unfolded}$. Under both noise models I and II, if $\sigma_1^2 = \tilde{\Omega}(\max(n^2, m))$, as $n \to \infty$, with high probability, we have $|\hat{\mathbf{x}}(j')| < |\hat{\mathbf{x}}(j)|$ for every $j$ in the bicluster and $j'$ outside of the bicluster.*

### 3.3 Thresholding Sum of Squared and Individual Trajectory Lengths

If the average trajectory lengths in the bicluster is larger than the one outside of the bicluster, methods based on trajectory length statistics can be successful in solving the tensor biclustering problem. One such method is thresholding individual trajectory lengths. In this method, we select $k^2$ indices $(j_1, j_2)$ with the largest trajectory length $\|\mathcal{T}(j_1, j_2, :)\|$ (SM Section 2).

**Theorem 3.** *As $n \to \infty$, with high probability, $\hat{J}_1 = J_1$ and $\hat{J}_2 = J_2$*

   - *if $\sigma_1^2 = \tilde{\Omega}(\sqrt{m}k^2)$, under noise model I.*

   - *if $\sigma_1^2 = \tilde{\Omega}(mk^2)$, under noise model II.*

Another method to solve the tensor biclustering problem is thresholding sum of squared trajectory lengths. In this method, we select $k$ row indices with the largest sum of squared trajectory lengths along the columns as an estimation of $J_1$. We estimate $J_2$ similarly (SM Section 2).

**Theorem 4.** *As $n \to \infty$, with high probability, $\hat{J}_1 = J_1$ and $\hat{J}_2 = J_2$*

   - *if $\sigma_1^2 = \tilde{\Omega}(k\sqrt{nm})$, under noise model I.*

   - *if $\sigma_1^2 = \tilde{\Omega}(mk^2 + k\sqrt{nm})$, under noise model II.*

## 4 Statistical (Information-Theoretic) Limits of the Tensor Biclustering Problem

### 4.1 Coherent Case

In this section, we study a statistical (information theoretic) boundary for the tensor biclustering problem under the following statistical model: We assume $\mathbf{u}_1(j_1) = 1/\sqrt{k}$ for $j_1 \in J_1$. Similarly, we assume $\mathbf{w}_1(j_2) = 1/\sqrt{k}$ for $j_2 \in J_2$. Moreover, we assume $\mathbf{v}_1$ is a fixed given vector with $\|\mathbf{v}_1\| = 1$. In the next section, we consider a non-coherent model where $\mathbf{v}_1$ is random and unknown.

Let $\mathcal{T}$ be an observed tensor from the tensor biclustering model $(J_1, J_2)$. Let $J_{all}$ be the set of all possible $(J_1, J_2)$. Thus, $|J_{all}| = \binom{n}{k}^2$. A maximum likelihood estimator (MLE) for the tensor biclustering problem can be written as:

$$\max_{\hat{J} \in J_{all}} \mathbf{v}_1^t \sum_{(j_1, j_2) \in \hat{J}_1 \times \hat{J}_2} \mathcal{T}(j_1, j_2, :) - \frac{k(1 - \sigma_z^2)}{2\sigma_1} \sum_{(j_1, j_2) \in \hat{J}_1 \times \hat{J}_2} \|\mathcal{T}(j_1, j_2, :)\|^2 \qquad (4)$$
$$(\hat{J}_1, \hat{J}_2) \in J_{all}.$$

Note that under the noise model I, the second term is zero. To solve this optimization, one needs to compute the likelihood function for $\binom{n}{k}^2$ possible bicluster indices. Thus, the computational complexity of the MLE is exponential in $n$.

**Theorem 5.** *Under noise model I, if $\sigma_1^2 = \tilde{\Omega}(k)$, as $n \to \infty$, with high probability, $(J_1, J_2)$ is the optimal solution of optimization (4). A similar result holds under noise model II if $mk = \Omega(\log(n/k))$.*

Next, we establish an upper bound on $\sigma_1^2$ under which no computational method can solve the tensor biclustering problem with vanishing probability of error. This upper bound indeed matches with the MLE achievability bound of Theorem 5 indicating its tightness.

**Theorem 6.** *Let $\mathcal{T}$ be an observed tensor from the tensor biclustering model with bicluster indices $(J_1, J_2)$. Let $A$ be an algorithm that uses $\mathcal{T}$ and computes $(\hat{J}_1, \hat{J}_2)$. Under noise model I, for any*

*fixed* $0 < \alpha < 1$, *if* $\sigma_1^2 < c_\alpha k \log(n/k)$, *as* $n \to \infty$, *we have*

$$\inf_{A \in AllAlg} \sup_{(J_1, J_2) \in J_{all}} \mathbb{P}\left[\hat{J}_1 \neq J_1 \text{ or } \hat{J}_2 \neq J_2\right] > 1 - \alpha - \frac{\log(2)}{2k \log(ne/k)}. \tag{5}$$

*A similar result holds under noise model II if* $mk = \Omega(\log(n/k))$.

## 4.2 Non-coherent Case

In this section we consider a similar setup to the one of Section 4.1 with the difference that $\mathbf{v}_1$ is assumed to be uniformly distributed over a unit sphere. For simplicity, in this section we only consider noise model I. The ML optimization in this setup can be written as follows:

$$\max_{\hat{J} \in J_{all}} \| \sum_{(j_1, j_2) \in \hat{J}_1 \times \hat{J}_2} \mathcal{T}(j_1, j_2, :)\|^2 \tag{6}$$

$$(\hat{J}_1, \hat{J}_2) \in J_{all}.$$

**Theorem 7.** *Under noise model I, if* $\sigma_1^2 = \tilde{\Omega}(\max(k, \sqrt{km}))$, *as* $n \to \infty$, *with high probability,* $(J_1, J_2)$ *is the optimal solution of optimization* (6).

If $k > \Omega(m)$, the achievability bound of Theorem 7 simplifies to the one of Theorem 5. In this case, using the result of Theorem 6, this bound is tight. If $k < \mathcal{O}(m)$, the achievability bound of Theorem 7 simplifies to $\tilde{\Omega}(\sqrt{mk})$ which is larger than the one of Theorem 5 (this is the price we pay for not knowing $\mathbf{v}_1$). In the following, we show that this bound is also tight.

To show the converse of Theorem 7, we consider the detection task which is presumably easier than the estimation task. Consider two probability distributions: (1) $\mathbb{P}_{\sigma_1}$ under which the observed tensor is $\mathcal{T} = \sigma_1 \mathbf{u}_1 \otimes \mathbf{w}_1 \otimes \mathbf{v}_1 + \mathcal{Z}$ where $J_1$ and $J_2$ have uniform distributions over $k$ subsets of $[n]$ and $\mathbf{v}_1$ is uniform over a unit sphere. (2) $\mathbb{P}_0$ under which the observed tensor is $\mathcal{T} = \mathcal{Z}$. Noise entries are i.i.d. normal. We need the following definition of contiguous distributions ([8]):

**Definition 2.** *For every* $n \in \mathbb{N}$, *let* $\mathbb{P}_{0,n}$ *and* $\mathbb{P}_{1,n}$ *be two probability measures on the same measure space. We say that the sequence* $(\mathbb{P}_{1,n})$ *is contiguous with respect to* $(\mathbb{P}_{0,n})$, *if, for any sequence of events* $A_n$, *we have*

$$\lim_{n \to \infty} \mathbb{P}_{0,n}(A_n) = 0 \Rightarrow \lim_{n \to \infty} \mathbb{P}_{1,n}(A_n) = 0. \tag{7}$$

**Theorem 8.** *If* $\sigma_1^2 < \tilde{\mathcal{O}}(\sqrt{mk})$, $\mathbb{P}_{\sigma_1}$ *is contiguous with respect to* $\mathbb{P}_0$.

This theorem with Lemma 2 of [8] establishes the converse of Theorem 7. The proof is based on bounding the second moment of the Radon-Nikodym derivative of $\mathbb{P}_{\sigma_1}$ with respect to $\mathbb{P}_0$ (SM Section 4.9).

## 5 Summary of Asymptotic Results

Table 1 summarizes asymptotic bounds for the case of $\Delta = 1/\sqrt{k}$ and $m = \Theta(n)$. For the MLE we consider the coherent model of Section 4.1. Also in Table 1 we summarize computational complexity of different tensor biclustering methods. We discuss analytical and empirical running time of these methods in SM Section 2.2.

Table 1: Comparative analysis of tensor biclustering methods. Results have been simplified for the case of $m = \Theta(n)$ and $\Delta = 1/\sqrt{k}$.

| Methods | $\sigma_1^2$, noise model I | $\sigma_1^2$, noise model II | Comp. Complexity |
|---|---|---|---|
| Tensor Folding+Spectral | $\tilde{\Omega}(n^{3/2})$ | $\tilde{\Omega}(n^{3/2})$ | $\mathcal{O}(n^4)$ |
| Tensor Unfolding+Spectral | $\tilde{\Omega}(n^2)$ | $\tilde{\Omega}(n^2)$ | $\mathcal{O}(n^3)$ |
| Th. Sum of Squared Trajectory Lengths | $\tilde{\Omega}(nk)$ | $\tilde{\Omega}(nk^2)$ | $\mathcal{O}(n^3)$ |
| Th. Individual Trajectory Lengths | $\tilde{\Omega}(k^2\sqrt{n})$ | $\tilde{\Omega}(nk^2)$ | $\mathcal{O}(n^3)$ |
| Maximum Likelihood | $\tilde{\Omega}(k)$ | $\tilde{\Omega}(k)$ | $\exp(n)$ |
| Statistical Lower Bound | $\tilde{\mathcal{O}}(k)$ | $\tilde{\mathcal{O}}(k)$ | - |

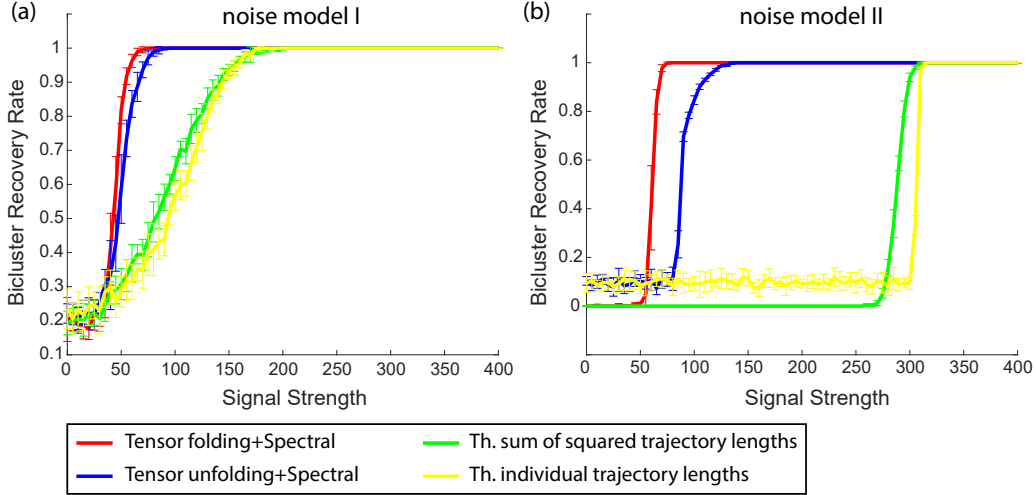

Figure 3: Performance of different tensor biclustering methods in various values of $\sigma_1$ (i.e., the signal strength), under both noise models I and II. We consider $n = 200$, $m = 50$, $k = 40$. Experiments have been repeated 10 times for each point.

In both noise models, the maximum likelihood estimator which has an exponential computational complexity leads to the best achievability bound compared to other methods. Below this bound, the inference is statistically impossible. Tensor folding+spectral method outperforms other methods with polynomial computational complexity if $k > \sqrt{n}$ under noise model I, and $k > n^{1/4}$ under noise model II. For smaller values of $k$, thresholding individual trajectory lengths lead to a better achievability bound. This case is a part of the high-SNR regime where the average trajectory lengths within the bicluster is significantly larger than the one outside of the bicluster. Unlike thresholding individual trajectory lengths, other methods use the entire tensor to solve the tensor biclustering problem. Thus, when $k$ is very small, the accumulated noise can dominate the signal strength. Moreover, the performance of the tensor unfolding method is always worst than the one of the tensor folding method. The reason is that, the tensor unfolding method merely infers a low dimensional subspace of trajectories, ignoring the block structure that true low dimensional trajectories form.

# 6   Numerical Results

## 6.1   Synthetic Data

In this section we evaluate the performance of different tensor biclustering methods in synthetic datasets. We use the statistical model described in Section 4.1 to generate the input tensor $\mathcal{T}$. Let $(\hat{J}_1, \hat{J}_2)$ be estimated bicluster indices $(J_1, J_2)$ where $|\hat{J}_1| = |\hat{J}_2| = k$. To evaluate the inference quality we compute the fraction of correctly recovered bicluster indices (SM Section 3.1).

In our simulations we consider $n = 200$, $m = 50$, $k = 40$. Figure 3 shows the performance of four tensor biclustering methods in different values of $\sigma_1$ (i.e., the signal strength), under both noise models I and II. Tensor folding+spectral algorithm outperforms other methods in both noise models. The gain is larger in the setup of noise model II compared to the one of noise model I.

## 6.2   Real Data

In this section we apply tensor biclustering methods to the roadmap epigenomics dataset [4] which provides histon mark signal strengths in different segments of human genome in various tissues and cell types. In this dataset, finding a subset of genome segments and a subset of tissues (cell-types) with highly correlated histon mark values can provide insight on tissue-specific functional roles of genome segments [4]. After pre-processing the data (SM Section 3.2), we obtain a data tensor $\mathcal{T} \in \mathbb{R}^{n_1 \times n_2 \times m}$ where $n_1 = 49$ is the number of tissues (cell-types), $n_2 = 1457$ is the number of

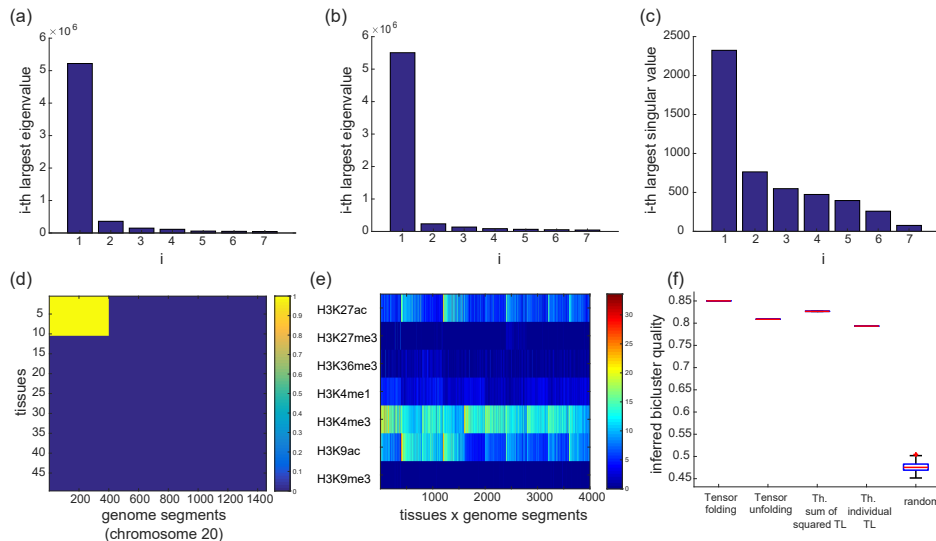

Figure 4: An application of tensor biclustering methods to the the roadmap epigenomics data.

genome segments, and $m = 7$ is the number of histon marks. Note that although in our analytical results for simplicity we assume $n_1 = n_2$, our proposed methods can be used in a more general case such as the one considered in this section.

We form two combined covariance matrices $\mathbf{C}_1 \in \mathbb{R}^{n_1 \times n_1}$ and $\mathbf{C}_2 \in \mathbb{R}^{n_2 \times n_2}$ according to (3). Figure 4-(a,b) shows largest eigenvalues of $\mathbf{C}_1$ and $\mathbf{C}_2$, respectively. As illustrated in these figures, spectral gaps (i.e., $\lambda_1 - \lambda_2$) of these matrices are large, indicating the existence of a low dimensional signal tensor in the input tensor. We also form an unfolded tensor $\mathbf{T}_{unfolded} \in \mathbb{R}^{m \times n_1 n_2}$. Similarly, there is a large gap between the first and second largest singular values of $\mathbf{T}_{unfolded}$ (Figure 4-c).

We use the tensor folding+spectral algorithm 1 with $|J_1| = 10$ and $|J_2| = 400$ (we consider other values for the bicluster size in SM Section 3.2). The output of the algorithm $(\hat{J}_1, \hat{J}_2)$ is illustrated in Figure 4-d (note that for visualization purposes, we re-order rows and columns to have the bicluster appear in the corner). Figure 4-e illustrates the unfolded subspace $\{\mathcal{T}(j_1, j_2, :) : (j_1, j_2) \in \hat{J}_1 \times \hat{J}_2\}$. In this inferred bicluster, Histon marks H3K4me3, H3K9ac, and H3K27ac have relatively high values. Reference [4] shows that these histon marks indicate a promoter region with an increased activation in the genome.

To evaluate the quality of the inferred bicluster, we compute total absolute pairwise correlations among vectors in the inferred bicluster. As illustrated in Figure 4-f, the quality of inferred bicluster by tensor folding+spectral algorithm is larger than the one of other methods. Next, we compute the bicluster quality by choosing bicluster indices uniformly at random with the same cardinality. We repeat this experiment 100 times. There is a significant gap between the quality of these random biclusters and the ones inferred by tensor biclustering methods indicating the significance of our inferred biclusters. For more details on these experiment, see SM Section 3.2.

## 7 Discussion

In this paper, we introduced and analyzed the tensor biclustering problem. The goal is to compute a subset of tensor rows and columns whose corresponding trajectories form a low dimensional subspace. To solve this problem, we proposed a method called tensor folding+spectral which demonstrated improved analytical and empirical performance compared to other considered methods. Moreover, we characterized computational and statistical (information theoretic) limits for the tensor biclustering problem in an asymptotic regime, under both coherent and non-coherent statistical models.

Our results consider the case when the rank of the subspace is equal to one (i.e., $q = 1$). When $q > 1$, in both tensor folding+spectral and tensor unfolding+spectral methods, the embedded subspace in the signal matrix will have a rank of $q > 1$, with singular values $\sigma_1 \geq \sigma_2 \geq ... \geq \sigma_q > 0$. In this

setup, we need the spectral radius of the noise matrix to be smaller than $\sigma_q$ in order to guarantee the recovery of the subspace. The procedure to characterize asymptotic achievability bounds would follow from similar steps of the rank one case with some technical differences. For example, we would need to extend Lemma 6 to the case where the signal matrix has rank $q > 1$. Moreover, in our problem setup, we assumed that the size of the bicluster $k$ and the rank of its subspace $q$ are know parameters. In practice, these parameters can be learned approximately from the data. For example, in the tensor folding+spectral method, a good choice for the $q$ parameter would be the index where eigenvalues of the folded matrix decrease significantly. Knowing $q$, one can determine the size of the bicluster similarly as the number of indices in top eigenvectors with significantly larger absolute values. Another practical approach to estimate model parameters would be trial and error plus cross validations.

Some of the developed proof techniques may be of independent interest as well. For example, we proved an $l_\infty$ version of the Davis-Kahan lemma for a Wishart noise matrix. Solving the tensor biclustering problem for the case of having multiple overlapping biclusters, for the case of having incomplete tensor, and for the case of a priori unknown bicluster sizes are among future directions.

# 8    Code

We provide code for tensor biclustering methods in the following link: `https://github.com/SoheilFeizi/Tensor-Biclustering`.

# 9    Acknowledgment

We thank Prof. Ofer Zeitouni for the helpful discussion on detectably proof techniques of probability measures.

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
