[Supplementary Material · Tensor Biclustering-SM.pdf]

# Supplementary Materials
# Tensor Biclustering

**Soheil Feizi**
Stanford University
sfeizi@stanford.edu

**Hamid Javadi**
Stanford University
hrhakim@stanford.edu

**David Tse**
Stanford University
dntse@stanford.edu

## Contents

## 1 Problem Formulation

### 1.1 Notation

In this document, we refer to pointers in the main text using the prefix *MT*. For example, equation MT-1 refers to equation 1 in the main text.

We use $\mathcal{T}$, $\mathcal{X}$, and $\mathcal{Z}$ to represent input, signal, and noise tensors. For matrices we use bold-faced upper case letters, for vectors we use bold-faced lower case letters, and for scalars we use regular lower case letters. For example, $\mathbf{X}$ represents a matrix, $\mathbf{x}$ represents a vector, and $x$ represents a scalar number. For any set $J$, $|J|$ denotes its cardinality. $\mathbf{I}_{n_1 \times n_2}$ and $\mathbf{1}_{n_1 \times n_2}$ are the identity and all one matrices of size $n_1 \times n_2$, respectively. When no confusion arises, we drop the subscripts. $[n]$ represents the set $\{1, 2, ..., n\}$. For $J \subset [n]$, $\mathbf{u}^{(J)}$ means that $u(j) = 0$ if $j \in J$. $\bar{J} = [n] - J$. $\mathcal{I}_{x=y}$ is the indicator function of the event $x = y$. $\mathbf{e}_i$ is a vector whose $i$-th entry is one and its other entries are zero. $X \overset{\mathrm{d}}{=} Y$ means random variables $X$ and $Y$ have the same distribution.

$Tr(\mathbf{X})$ and $\mathbf{X}^t$ represent the trace and the transpose of the matrix $\mathbf{X}$, respectively. $diag(\mathbf{x})$ is a diagonal matrix whose diagonal elements are equal to $\mathbf{x}$, while $diag(\mathbf{X})$ is a vector of the diagonal elements of the matrix $\mathbf{X}$. $\|\mathbf{x}\|_2 = (\mathbf{x}^t \mathbf{x})^{1/2}$ is the second norm of the vector $\mathbf{x}$. When no confusion arises, we drop the subscript. $\|\mathbf{x}\|_\infty$ is the infinity norm of the vector $\mathbf{x}$ (i.e., $\|\mathbf{x}\|_\infty = \max(|x_i|)$). $\|\mathbf{X}\|$ is the operator norm of the matrix $\mathbf{X}$, while $\|\mathbf{X}\|_F$ is its Frobenius norm. $< \mathbf{x}, \mathbf{y} >$ is the inner product between vectors $\mathbf{x}$ and $\mathbf{y}$. $\mathbf{x} \perp \mathbf{y}$ indicates that vectors $\mathbf{x}$ and $\mathbf{y}$ are orthogonal. The matrix inner product is defined as $< \mathbf{X}, \mathbf{Y} > = Tr(\mathbf{X}\mathbf{Y}^t)$. $\|\mathbf{X}\|_F^2 = < \mathbf{X}, \mathbf{X} >$. Inner product and Frobenius norm of a tensor are defined similarly. $\det(\mathbf{X})$ is the determinant of $\mathbf{X}$. $\mathbf{X} \otimes \mathbf{Y}$ indicates kronecker product of matrices $\mathbf{X}$ and $\mathbf{Y}$. $D_{KL}(\mathcal{P}_1 \| \mathcal{P}_2)$ represents the Kullback–Leibler (KL) divergence between two distributions $\mathcal{P}_1$ and $\mathcal{P}_2$.

The asymptotic notation $a(n) = \mathcal{O}(b(n))$ means that, there exists a universal constant $c$ such that for sufficiently large $n$, we have $|a(n)| < cb(n)$. If there exists $c > 0$ such that $a(n) = \mathcal{O}(b(n)\log(n)^c)$, we use the notation $a(n) = \tilde{\mathcal{O}}(b(n))$. The asymptotic notation $a(n) = \Omega(b(n))$ and $a(n) = \tilde{\Omega}(b(n))$ is the same as $b(n) = \mathcal{O}(a(n))$ and $b(n) = \tilde{\mathcal{O}}(a(n))$, respectively. Moreover, we write $a(n) = \Theta(b(n))$ iff $a(n) = \Omega(b(n))$ and $b(n) = \Omega(a(n))$. Similarly, we write $a(n) = \tilde{\Theta}(b(n))$ iff $a(n) = \tilde{\Omega}(b(n))$ and $b(n) = \tilde{\Omega}(a(n))$.

## 1.2  Signal and Noise Models

Consider $q = 1$ in MT-(1), the tensor biclustering model simplifies to

$$\mathcal{T} = \mathcal{X} + \mathcal{Z} = \sigma_1 \mathbf{u}_1 \mathbf{w}_1 \mathbf{v}_r + \mathcal{Z}, \tag{1}$$

In this section, we explain noise models I and II with more details:

- **Noise Model I:** In this model, the variance of the noise within and outside of bicluster indices is assumed to be the same. Thus, under this model we have

$$\sigma_z^2 = 1. \tag{2}$$

  This is the noise model often considered in analysis of sub-matrix localization [1, 2] and tensor PCA [3, 4, 5, 6, 7, 8, 9]. Although this model simplifies the analysis, it has the following drawback: under this noise model, for every value of $\sigma_1^2$, the average trajectory length within the bicluster is larger than the average trajectory length outside of the bicluster. To see this, let $\mathbf{T}_1 \in \mathbb{R}^{m \times k^2}$ be a matrix whose columns include trajectories $\mathcal{T}(j_1, j_2, :)$ for $(j_1, j_2) \in J_1 \times J_2$ (i.e., $\mathbf{T}_1$ is the unfolded $\mathcal{T}(J_1, J_2, :)$). We can write $\mathbf{T}_1 = \mathbf{X}_1 + \mathbf{Z}_1$ where $\mathbf{X}_1$ and $\mathbf{Z}_1$ are unfolded $\mathcal{X}(J_1, J_2, :)$ and $\mathcal{Z}(J_1, J_2, :)$, respectively. The squared Frobenius norm of $\mathbf{X}_1$ is equal to $\|\mathbf{X}_1\|_F^2 = \sigma_1^2$. Moreover, the squared Frobenius norm of $\mathbf{Z}_1$ has a $\chi$-squared distribution with $mk^2$ degrees of freedom. Thus, the average squared Frobenius norm of $\mathbf{T}_1$ is equal to $\sigma_1^2 + \sigma_z^2 mk^2$. Let $\mathbf{T}_2 \in \mathbb{R}^{m \times k^2}$ be a matrix whose columns include only noise trajectories. Using a similar argument, we have $\mathbb{E}[\|\mathbf{T}_2\|_F^2] = mk^2$, which is smaller than $\sigma_1^2 + mk^2$.

- **Noise Model II:** In this model, $\sigma_z^2$ is modeled to minimize the difference between the average trajectory lengths within and outside of the bicluster. If $\sigma_1^2 < mk^2$, without noise, the average trajectory lengths in the bicluster is smaller than the one outside of the bicluster. In this regime, having $\sigma_z^2 = 1 - \sigma_1^2/mk^2$ makes the average trajectory lengths within and outside of the bicluster comparable. This regime is called the low-SNR regime. If $\sigma_1^2 > mk^2$, the average trajectory lengths in the bicluster is larger than the one outside of the bicluster. This regime is called the high-SNR regime. In this regime, adding noise to signal trajectories increases their lengths and makes solving the tensor biclustering problem easier. Therefore, in this regime we assume $\sigma_z^2 = 0$

to minimize the difference between average trajectory lengths within and outside of the bicluster. Therefore, under the noise model II, we have

$$\sigma_z^2 = \max(0, 1 - \frac{\sigma_1^2}{mk^2}). \qquad (3)$$

## 1.3 Related Work

A related model to tensor biclustering (1) is the spiked tensor model [3]:

$$\mathcal{T} = \sigma_1 \mathbf{v} \otimes \mathbf{v} \otimes \mathbf{v} + \mathcal{Z}. \qquad (4)$$

Unlike the tensor biclustering model which is asymmetric along tensor dimensions, the spiked tensor model has a symmetric structure. Assuming the noise tensor $\mathcal{Z}$ has i.i.d. standard normal entries, reference [4] has shown that, if $\sigma_1^2 < (1-\epsilon)n$, no algorithm based on a spectral statistical test can detect the existence of such signal structure, with error probability vanishing as $n \to \infty$. References [6, 5] have shown that the inference of the signal tensor is possible using a polynomial time algorithm if $\sigma_1^2 = \tilde{\Omega}(n^{3/2})$. A variation of this bound also appears in the analysis of our tensor folding method (Theorem MT-1). Moreover, statistical and computational trade-offs of a generalized tensor PCA model have been studied in [9].

In the model (1), if $m = 1$, the tensor can be viewed as a matrix. Let $\mathbf{u}_1(j_1) = 1/\sqrt{k}$ and $\mathbf{w}_1(j_2) = 1/\sqrt{k}$ for $(j_1, j_2) \in J_1 \times J_2$, and consider noise model I. In this case, elements of the sub-matrix $\mathcal{T}(J_1, J_2, 1)$ have i.i.d. Gaussian distributions with $\mu = \sigma_1/k$ means and unit variances. Elements of the matrix outside of indices $J_1 \times J_2$ have normal distributions. In this special case, the tensor biclustering problem simplifies to the sub-matrix localization problem [1, 2]. Note that the bicluster structure in this special case comes from scaling coefficients $\mathbf{u}_1$ and $\mathbf{w}_1$ since $\mathbf{v}_1 = 1$. This problem is closely related to the planted clique, bi-clustering (co-clustering), and community detection problems. In this case, our statistical lower bound (Theorem MT-6) and the achievability bound of the MLE (Theorem MT-5) match with the ones derived specifically for the sub-matrix localization problem [1, 2].

## 2 Details of Tensor Biclustering Methods

### 2.1 Algorithms

In this section, we provide more details on three tensor biclustering methods, namely tensor unfolding+spectral, thresholding sum of squared trajectory lengths, and thresholding individual trajectory lengths.

---

**Algorithm 1** Tensor Unfolding+Spectral

---

**Input:** $\mathcal{T}, k$
Compute $\hat{\mathbf{x}}$, the top right singular vector of $\mathbf{T}_{unfolded}$
Let $\hat{J}_1$ be the set of tensor row indices of $k^2$ largest entries of $|\hat{\mathbf{x}}|$
Let $\hat{J}_2$ be the set of tensor column indices of $k^2$ largest entries of $|\hat{\mathbf{x}}|$
**Output:** $\hat{J}_1$ and $\hat{J}_2$

---

**Algorithm 2** Thresholding Individual Trajectory Lengths

---

**Input:** $\mathcal{T}, k$
Compute $\hat{J}_1$, the set of first indices of $k^2$ largest trajectories
Compute $\hat{J}_2$, the set of second indices of $k^2$ largest trajectories
**Output:** $\hat{J}_1$ and $\hat{J}_2$

---

**Algorithm 3** Thresholding Sum of Squared Trajectory Lengths

---

**Input:** $\mathcal{T}, k$
Compute $\mathbf{d}_1(j_1) = \sum_{j_2=1}^{n} \|\mathcal{T}(j_1, j_2, :)\|^2$ for $1 \leq j_1 \leq n$
Compute $\mathbf{d}_2(j_2) = \sum_{j_1=1}^{n} \|\mathcal{T}(j_1, j_2, :)\|^2$ for $1 \leq j_2 \leq n$
Compute $\hat{J}_1$, the index set of $k$ largest components of $\mathbf{d}_1$
Compute $\hat{J}_2$, the index set of $k$ largest components of $\mathbf{d}_2$
**Output:** $\hat{J}_1$ and $\hat{J}_2$

---

## 2.2 Computational Complexity

In Table MT-1 we summarize computational complexity of different tensor biclustering methods. Tensor unfolding+spectral, thresholding sum of squared trajectory lengths, and thresholding individual trajectory lengths have linear computational complexity with respect to the tensor size $n^2 m$. Computational complexity of the tensor folding+spectral is $\mathcal{O}(n^2 m^2)$ which is higher than linear and lower than quadratic with respect to the tensor size. Computational complexity of the MLE method is exponential in $n$.

Figure 1 shows the empirical running time of different tensor biclustering methods with respect to the tensor size $N = n^2 m$. In Figure 1-a, we vary the tensor size by varying $m$, while in Figure 1-b we increase the tensor size by increasing the size of all dimensions. In both setups, tensor unfolding+spectral method has the worst empirical running time compared to other methods. In the setup of panel (a), tensor folding+spectral method has a larger running time compared to thresholding individual and sum of squared trajectory lengths since its computational complexity depends on $m^2$ while the computational complexity of other methods depend on $m$. Our empirical running time analysis has been performed on an ordinary laptop using implementations of tensor biclustering methods in MATLAB.

## 3 Details of Numerical Experiments

### 3.1 Synthetic Data

In Section MT-6.1, we evaluate the performance of different tensor biclustering methods in synthetic datasets. We use the statistical model described in Section MT-4.1 to generate the input tensor $\mathcal{T}$. Let $(\hat{J}_1, \hat{J}_2)$ be estimated bicluster indices $(J_1, J_2)$ where $|\hat{J}_1| = |\hat{J}_2| = k$. To evaluate the inference quality we compute the following score:

$$\frac{|\hat{J}_1 \cap J_1|}{2k} + \frac{|\hat{J}_2 \cap J_2|}{2k}. \tag{5}$$

This score is always between zero and one. If $(\hat{J}_1, \hat{J}_2) = (J_1, J_2)$, this score achieves its maximum value one.

Tensor unfolding+spectral method (Algorithm 1) and thresholding individual trajectory lengths method (Algorithm 2) may have an output $(\hat{J}_1, \hat{J}_2)$ where $|\hat{J}_1| > k$ or $|\hat{J}_2| > k$. This is because these algorithms ignore the block structure formed by bicluster indices. To have a fair comparison with other methods, we select $k$ most repeated indices in their outputs as an estimate of bicluster indices.

### 3.2 Real Data

In Section MT-6.2, we apply different tensor biclustering methods to the roadmap epigenomics dataset [10] which provides histon mark signal strengths in different segments of human genome in various tissues and cell types. This dataset can be viewed as a three dimensional tensor whose dimensions represent segments of genome, tissues (cell types), and histon marks. Reference [10] has shown that in a tissue, segments of genome with similar histon mark values are often have similar functional roles (e.g., they are enhancers, promoters, etc.). Moreover, histon marks of a specific genome segment can vary across different tissues and cell-types.

Here we consider a portion of the roadmap epigenomics dataset to demonstrate applicability of tensor biclustering methods to this data type. A full analysis of the roadmap epigenomics dataset along

Figure 1: Empirical running time of different tensor biclustering methods with respect to the tensor size $n^2 m$. In panel (a) we consider $n = 40$, $m = \alpha 40$ and vary $\alpha$, while in panel (b) we consider $n = m = \alpha^{1/3} 40$. Experiments have been repeated 10 times for each point.

with biological validations of inferences are beyond the scope of the present paper. We consider genome segments of chromosome 20 in human. Each segment has 1000 base pairs. In each genome segment, we consider the average value of the signal strength for every histon mark. We only consider segments with at least one non-zero histon mark value. In the roadmap epigenomics dataset, some tissues (cell-types) do not have data for some histon marks. Thus, we only consider a subset of tissues (cell-types) and a subset of histon marks with complete data. After these filtering steps, we obtain a data tensor $\mathcal{T} \in \mathbb{R}^{n_1 \times n_2 \times m}$ where $n_1 = 49$ is the number of tissues (cell-types), $n_2 = 1457$ is the number of genome segments, and $m = 7$ is the number of histon marks. Our seven histon marks include the core set of five histone modification marks reported in reference [10] (i.e., H3K4me3, H3K4me1, H3K36me3, H3K27me3, and H3K9me3), along with two additional marks (i.e., H3K27ac and H3K9ac). $\mathcal{T}(j_1, j_2, i)$ provides the signal strength of histon mark $i$ in the genome segment $j_2$ of tissue $j_1$. Our goal is to find $J_1 \subset [n_1]$ and $J_2 \subset [n_2]$ where $\{\mathcal{T}(j_1, j_2, :) : (j_1, j_2) \in J_1 \times J_2\}$ form a low dimensional subspace.

To evaluate the quality of the inferred bicluster, we compute the total absolute pairwise correlations among vectors in the inferred bicluster, i.e.,

$$\frac{\sum_{(j_1, j_2) \neq (j_1', j_2') \in \hat{J}_1 \times \hat{J}_2} |\rho(\mathcal{T}(j_1, j_2, :), \mathcal{T}(j_1', j_2', :))|}{(|\hat{J}_1||\hat{J}_2|)^2 - |\hat{J}_1||\hat{J}_2|} \tag{6}$$

where $\rho(., .)$ indicates the Pearson's correlation between two vectors. If all vectors in the inferred bicluster are parallel to each other, this value will be one. If vectors in the inferred bicluster are orthogonal to each other, this value will be zero.

We evaluate the quality of inferred biclusters for different cluster sizes in Figure 2. Similar to the setup considered in the main text, in these cases the tensor folding+spectral method continues to outperform other tensor biclustering methods.

## 4  Proofs

### 4.1  Preliminary Lemmas

For a sub-Gaussian variable $X$, $\|X\|_{\psi_2}$ denotes the sub-Gaussian norm of $X$ defined as

$$\|X\|_{\psi_2} \triangleq \sup_{p \geq 1} p^{-\frac{1}{2}} (\mathbb{E}[|X|^p])^{1/p}. \tag{7}$$

If $X$ is a centered Gaussian variable with variance $\sigma^2$, then $\|X\|_{\psi_2} \leq c\sigma$ where $c$ is an absolute constant.

Figure 2: The quality of inferred biclusters by different tensor biclustering methods and uniformly randomly selected bicluster indices when (a) $|\hat{J}_1| = 5$, $|\hat{J}_2| = 200$, and (b) $|\hat{J}_1| = 20$ and $|\hat{J}_2| = 800$.

**Lemma 1.** *Let $X_1,...,X_N$ be independent, centered sub-Gaussian random variables. Then, for every $\mathbf{a} = (a_1,...,a_N)^T \in \mathbb{R}^N$ and every $t \geq 0$, we have*

$$\mathbb{P}\left[\left|\sum_{i=1}^{N} a_i X_i\right| > t\right] < e\exp\left(-\frac{ct^2}{\sigma_m^2\|\mathbf{a}\|_2^2}\right), \tag{8}$$

*where $\sigma_m = \max_i \|X_i\|_{\psi_2}$ and $c > 0$ is an absolute constant.*

**Proof.** See Proposition 5.10 in [11].

Let $Y$ be a sub-exponential random variable. $\|Y\|_{\psi_1}$ denotes the sub-exponential norm of $Y$, defined as

$$\|Y\|_{\psi_1} \triangleq \sup_{p\geq 1} p^{-1}(\mathbb{E}[|Y|^p])^{1/p}. \tag{9}$$

If $X$ is sub-Gaussian, $Y = X^2$ is sub-exponential, and vice versa. Moreover, we have

$$\|X\|_{\psi_2}^2 \leq \|Y\|_{\psi_1} \leq 2\|X\|_{\psi_2}^2. \tag{10}$$

**Lemma 2.** *Let $Y_1,...,Y_N$ be independent, centered sub-exponential random variables. Then, for every $\mathbf{a} = (a_1,...,a_N)^T \in \mathbb{R}^N$ and every $t \geq 0$, we have*

$$\mathbb{P}\left[\left|\sum_{i=1}^{N} a_i Y_i\right| > t\right] < 2\exp\left[-c\min\left(\frac{t^2}{\sigma_m^2\|\mathbf{a}\|_2^2}, \frac{t}{\sigma_m\|\mathbf{a}\|_\infty}\right)\right], \tag{11}$$

*where $\sigma_m = \max_i \|Y_i\|_{\psi_1}$ and $c > 0$ is an absolute constant.*

**Proof.** See Proposition 5.16 in [11].

To bound the operator norm of sum of random matrices we use the following lemma:

**Lemma 3.** *Let $\mathbf{Y}_1,...,\mathbf{Y}_N$ be $n_1 \times n_2$ independent random matrices such that for all $j \in [N]$*

$$\mathbb{P}\left[\|\mathbf{Y}_j - \mathbb{E}[\mathbf{Y}_j]\| \geq \beta\right] \leq p_1. \tag{12}$$

*Moreover suppose we have*

$$\left\|\mathbb{E}[\mathbf{Y}_j] - \mathbb{E}\left[\mathbf{Y}_j \mathcal{I}_{\|\mathbf{Y}_j\|<\beta}\right]\right\| \leq p_2. \tag{13}$$

*Let*

$$\mu^2 = \max\left(\left\|\sum_{j=1}^{N} \mathbb{E}[\mathbf{Y}_j \mathbf{Y}_j^t] - \mathbb{E}[\mathbf{Y}_j]\mathbb{E}[\mathbf{Y}_j^t]\right\|, \left\|\sum_{j=1}^{N} \mathbb{E}[\mathbf{Y}_j^t \mathbf{Y}_j] - \mathbb{E}[\mathbf{Y}_j^t]\mathbb{E}[\mathbf{Y}_j]\right\|\right) \tag{14}$$

*Then for $\mathbf{Y} = \sum_{j=1}^{N} \mathbf{Y}_j$, we have*

$$\mathbb{P}\left[\|\mathbf{Y} - \mathbb{E}[\mathbf{Y}]\| \geq t\right] \leq Np_1 + (n_1 + n_2)\exp\left(\frac{-(t - Np_2)^2}{2(\mu^2 + \beta(t - Np_2)/3)}\right). \tag{15}$$

**Proof.** See Proposition A.7 in [5].

**Lemma 4.** *Let* $\mathbf{Z}_1,...,\mathbf{Z}_N$ *be* $m \times n$ *independent random matrices such that* $\mathbf{Z}_r(i,j)$ *has a standard normal distribution for every* $i,j$. *Then, for some constant* $c > 0$, *with high probability, we have*

$$\left\|\sum_{j=1}^{N}\mathbf{Z}_j^t\mathbf{Z}_j - Nm\mathbf{I}\right\| < c\max(n,m)\sqrt{N}\log(N). \tag{16}$$

**Proof.** Let $n_1 = max(n,m)$. Let $\mathbf{Y}_j = \mathbf{Z}_j^t\mathbf{Z}_j$. We have $\|\mathbf{Y}_j\| = \|\mathbf{Z}_j\|^2$. Since $\|\mathbf{Z}_j\|$ has a sub-Gaussian tail distribution, for some constant $c > 0$, we have

$$\mathbb{P}\left[\|\mathbf{Z}_j\| > \sqrt{tn_1}\right] \leq \exp(-ct). \tag{17}$$

Therefore, we have

$$\mathbb{P}\left[\|\mathbf{Y}_j - \mathbb{E}[\mathbf{Y}_j]\| > (t+1)n_1\right] \leq \mathbb{P}\left[\|\mathbf{Y}_j\| > tn_1\right] \leq \exp(-ct). \tag{18}$$

Moreover, we have

$$\left\|\mathbb{E}\left[\mathbf{Y}_j\right] - \mathbb{E}\left[\mathbf{Y}_j\mathcal{I}_{\|\mathbf{Y}_j\|<\beta}\right]\right\| = \mathbb{P}\left[\|\mathbf{Y}_j\| > \beta\right]\|\mathbb{E}[\mathbf{Y}_j]\| = m\exp\left(\frac{-c\beta}{n_1}\right) < n_1\exp\left(\frac{-c\beta}{n_1}\right). \tag{19}$$

Thus, for a large enough constant $c' > 0$ and having $\beta = c'n_1\log(n)$ in (18) and (19), we can satisfy conditions (12) and (13) of Lemma 3 for $p_1 = p_2 = n^{-c_2}$, for a sufficiently large $c_2 > 0$.

Next we compute $\mu^2$ in (14). Since $\mathbf{Y}_j$ is symmetric we have

$$\mu^2 = \left\|\sum_{j=1}^{N}\mathbb{E}[\mathbf{Y}_j\mathbf{Y}_j^t] - \mathbb{E}[\mathbf{Y}_j]\mathbb{E}[\mathbf{Y}_j^t]\right\| \tag{20}$$

$$\leq \left\|\sum_{j=1}^{N}\mathbb{E}[\mathbf{Y}_j\mathbf{Y}_j^t]\right\| + Nm^2$$

$$\leq \sum_{j=1}^{N}\mathbb{E}[\|\mathbf{Y}_j\mathbf{Y}_j^t\|] + Nm^2,$$

where the last step follows form Jenson's inequality. Moreover, we have

$$\mathbb{E}[\|\mathbf{Y}_j\mathbf{Y}_j^t\|] = \int_0^\infty \mathbb{P}\left[\|\mathbf{Y}_j\mathbf{Y}_j^t\| > t\right]dt \tag{21}$$

$$= \int_0^\infty \mathbb{P}\left[\|\mathbf{Z}_j\| > t^{1/4}\right]dt$$

$$= \int_0^\infty \exp\left(\frac{-c\sqrt{t}}{n_1}\right) = \frac{2n_1^2}{c^2}$$

Therefore, with high probability, $\mu^2 = \tilde{\mathcal{O}}(Nn_1^2)$. Substituting $p_1$, $p_2$ and $\mu^2$ in (15) completes the proof of this lemma.

**Lemma 5.** *Let* $\mathbf{x} = (x_1, x_2, \ldots, x_n) \in \mathbb{R}^n$ *be a random vector with independent sub-Gaussian entries* $x_i$ *with* $\mathbb{E}x_i = 0$ *and* $\|x_i\|_{\psi_2} \leq K$. *For* $\mathbf{A} \in \mathbb{R}^{n\times n}$, $t \geq 0$, *we have*

$$\mathbb{P}\{|\langle\mathbf{x}, \mathbf{A}\mathbf{x}\rangle - \mathbb{E}\langle\mathbf{x}, \mathbf{A}\mathbf{x}\rangle|\} \leq 2\exp\left\{-c\min\left(\frac{t^2}{K^4\|\mathbf{A}\|_F^2}, \frac{t}{K^2\|\mathbf{A}\|}\right)\right\} \tag{22}$$

*for some constant* $c > 0$.

**Proof.** See reference [12].

**Lemma 6.** *Let* $\mathbf{Y} = \mathbf{x}\mathbf{x}^T + \sigma\mathbf{W}$, *where* $\mathbf{x} \in \mathbb{R}^n, \|\mathbf{x}\|_2 = 1$ *and* $\mathbf{W} \in \mathbb{R}^{n\times n} = \sum_{j=1}^{N}\left(\mathbf{z}_j\mathbf{z}_j^T - \mathbb{E}\mathbf{z}_j\mathbf{z}_j^T\right)$ *where* $\mathbf{z}_j$ *are i.i.d.* $\mathcal{N}(0, \mathbf{I}_{n\times n})$ *gaussian random vectors. Let* $\widetilde{\mathbf{x}} \in \mathbb{R}^n$,

$\|\widetilde{\mathbf{x}}\|_2 = 1$ *be the eigenvector corresponding to the largest eigenvalue of the matrix* $\mathbf{Y}$. *Let the operator norm of the matrix* $\mathbf{W}$ *be such that*

$$\|\mathbf{W}\|_2 \leq \lambda_{n,N}, \tag{23}$$

*with probability at least* $1 - O(n^{-2})$. *Further, let*

$$\sigma \leq \frac{c_0}{\lambda_{n,N}}, \tag{24}$$

*for some positive constant* $c_0 < 1/6$. *Letting* $M = \|\mathbf{x}\|_\infty$, *we have*

$$\|\widetilde{\mathbf{x}} - \mathbf{x}\|_\infty \leq \mathcal{O}\left(\sigma\left(\sqrt{N \log N} + M\lambda_{n,N}\right)\right), \tag{25}$$

*with high probability as* $n \to \infty$.

**Proof.** Our proof is based on the technique used in [13]. However, in [13], $\mathbf{x} \in \mathbb{C}^n$, $\mathbf{W} \in \mathbb{C}^{n \times n}$ and $|x_i| = 1$ for $1 \leq i \leq n$. In addition, in [13], it is assumed that $\mathbf{W}$ is a complex Wigner random matrix which is not the case in our model. Hence, we will develop a bound that fits our model. For $1 \leq l \leq n$, we denote the $l$-th row (or column) of $\mathbf{W}$ by $\mathbf{w}_l$. Further, we define $\mathbf{W}^{(l)} \in \mathbb{R}^{n \times n}$ as

$$W_{i,j}^{(l)} \triangleq W_{i,j}, \quad \text{for} \quad i \neq l, j \neq l, \tag{26}$$
$$W_{i,j}^{(l)} \triangleq 0, \quad \text{if} \quad i = l \text{ or } j = l.$$

Further, we define $\Delta \mathbf{W}^{(l)} \triangleq \mathbf{W} - \mathbf{W}^{(l)}$. Note that (23) results in

$$\left\|\mathbf{W}^{(l)}\right\|_2 \leq \lambda_{n,N}, \qquad \left\|\Delta \mathbf{W}^{(l)}\right\|_2 \leq \lambda_{n,N}, \qquad \|\mathbf{w}_l\|_2 \leq \lambda_{n,N}, \tag{27}$$

with probability at least $1 - O(n^{-2})$. Let $\widetilde{\mathbf{x}}^{(l)}$ be the eigenvector corresponding to the top eigenvalue of the matrix $\mathbf{Y}^{(l)} = \mathbf{x}\mathbf{x}^T + \mathbf{W}^{(l)}$. Note that for any $1 \leq l \leq n$, we can write

$$|\tilde{x}_l - x_l| = \left|\frac{(\mathbf{Y}\widetilde{\mathbf{x}})_l}{\lambda_1(\mathbf{Y})} - x_l\right| = \left|\frac{\left(\mathbf{x}\mathbf{x}^T\widetilde{\mathbf{x}}\right)_l + \sigma(\mathbf{W}\widetilde{\mathbf{x}})_l}{\lambda_1(\mathbf{Y})} - x_l\right| \tag{28}$$
$$\leq \left|\frac{|\langle \mathbf{x}, \widetilde{\mathbf{x}}\rangle|}{\lambda_1(\mathbf{Y})} - 1\right| M + \frac{\sigma |(\mathbf{W}\widetilde{\mathbf{x}})_l|}{\lambda_1(\mathbf{Y})}.$$

Hence,

$$\|\widetilde{\mathbf{x}} - \mathbf{x}\|_\infty \leq \left|\frac{|\langle \mathbf{x}, \widetilde{\mathbf{x}}\rangle|}{\lambda_1(\mathbf{Y})} - 1\right| M + \frac{\sigma \|\mathbf{W}\widetilde{\mathbf{x}}\|_\infty}{\lambda_1(\mathbf{Y})}. \tag{29}$$

Next we bound the term $\|\mathbf{W}\widetilde{\mathbf{x}}\|_\infty$. Note that for $1 \leq l \leq n$, we have

$$|(\mathbf{W}\widetilde{\mathbf{x}})_l| = |\langle \mathbf{w}_l, \widetilde{\mathbf{x}}\rangle| = \left|\left\langle \mathbf{w}_l, \widetilde{\mathbf{x}}^{(l)}\right\rangle\right| + \left|\left\langle \mathbf{w}_l, \widetilde{\mathbf{x}} - \widetilde{\mathbf{x}}^{(l)}\right\rangle\right| \leq \left|\left\langle \mathbf{w}_l, \widetilde{\mathbf{x}}^{(l)}\right\rangle\right| + \|\mathbf{w}_l\|_2 \left\|\widetilde{\mathbf{x}} - \widetilde{\mathbf{x}}^{(l)}\right\|_2. \tag{30}$$

Note that $\mathbf{Y} = \mathbf{Y}^{(l)} + \sigma\Delta \mathbf{W}^{(l)}$. Hence, using the Davis-Kahan *sin* $\Theta$ Theorem (see, e.g., Lemma 11 in [13]), we have

$$\left\|\widetilde{\mathbf{x}} - \widetilde{\mathbf{x}}^{(l)}\right\|_2 \leq \frac{\sigma\sqrt{2}\left\|\Delta \mathbf{W}^{(l)}\widetilde{\mathbf{x}}^{(l)}\right\|_2}{\delta\left(\mathbf{Y}^{(l)}\right) - \sigma\left\|\Delta \mathbf{W}^{(l)}\right\|_2}, \tag{31}$$

where $\delta\left(\mathbf{Y}^{(l)}\right) = \lambda_1(\mathbf{Y}^{(l)}) - \lambda_2(\mathbf{Y}^{(l)})$ is the spectral gap of the matrix $\mathbf{Y}^{(l)}$. Note that using the Weyl's inequality we can write

$$\delta\left(\mathbf{Y}^{(l)}\right) \geq \delta\left(\mathbf{x}\mathbf{x}^T\right) - 2\sigma\left\|\mathbf{W}^{(l)}\right\|_2 \geq 1 - 2\sigma\lambda_{n,N}, \tag{32}$$

where we have used (27) in the last inequality. Therefore, using (31), (27), (24) we get

$$\left\|\widetilde{\mathbf{x}} - \widetilde{\mathbf{x}}^{(l)}\right\|_2 \leq \frac{\sigma\sqrt{2}}{1 - 3\sigma\lambda_{n,N}}\left\|\Delta \mathbf{W}^{(l)}\widetilde{\mathbf{x}}^{(l)}\right\|_2 \leq \frac{\sqrt{2}}{3\lambda_{n,N}}\left\|\Delta \mathbf{W}^{(l)}\widetilde{\mathbf{x}}^{(l)}\right\|_2, \tag{33}$$

with probability at least $1 - O(n^{-2})$. Putting this in (30) we get,

$$|(\mathbf{W}\widetilde{\mathbf{x}})_l| \leq \left|\left\langle \mathbf{w}_l, \widetilde{\mathbf{x}}^{(l)} \right\rangle\right| + \frac{\sqrt{2}\,\|\mathbf{w}_l\|_2}{3\lambda_{n,N}}\left\|\Delta\mathbf{W}^{(l)}\widetilde{\mathbf{x}}^{(l)}\right\|_2, \tag{34}$$

with probability at least $1 - O(n^{-2})$. Note that $\left\|\Delta\mathbf{W}^{(l)}\widetilde{\mathbf{x}}^{(l)}\right\|_2 \geq \left|\left\langle \mathbf{w}_l, \widetilde{\mathbf{x}}^{(l)} \right\rangle\right|$. Hence, by (27), we get

$$|(\mathbf{W}\widetilde{\mathbf{x}})_l| \lesssim \left\|\Delta\mathbf{W}^{(l)}\widetilde{\mathbf{x}}^{(l)}\right\|_2, \tag{35}$$

with probability at least $1 - O(n^{-2})$. Thus, we need to bound the term $\left\|\Delta\mathbf{W}^{(l)}\widetilde{\mathbf{x}}^{(l)}\right\|_2$. Note that here we can leverage the independence between $\Delta\mathbf{W}^{(l)}$ and $\widetilde{\mathbf{x}}^{(l)}$ to get a tight bound on $\|\widetilde{\mathbf{x}} - \mathbf{x}\|_\infty$. We can write

$$\begin{aligned}
\left\|\Delta\mathbf{W}^{(l)}\widetilde{\mathbf{x}}^{(l)}\right\|_2^2 &= \left|\left(\Delta\mathbf{W}^{(l)}\widetilde{\mathbf{x}}^{(l)}\right)_l\right|^2 + \sum_{k\neq l}\left|\left(\Delta\mathbf{W}^{(l)}\widetilde{\mathbf{x}}^{(l)}\right)_k^2\right| \\
&\leq \left\langle \mathbf{w}_l, \widetilde{\mathbf{x}}^{(l)} \right\rangle^2 + \left|\widetilde{\mathbf{x}}_l^{(l)}\right|^2 \|\mathbf{w}_l\|_2^2 \leq \left\langle \mathbf{w}_l, \widetilde{\mathbf{x}}^{(l)} \right\rangle^2 + \left|\widetilde{\mathbf{x}}_l^{(l)}\right|^2 \lambda_{n,N}^2,
\end{aligned} \tag{36}$$

with probability at least $1 - O(n^{-2})$. In order to bound the term $\left\langle \mathbf{w}_l, \widetilde{\mathbf{x}}^{(l)} \right\rangle$, it suffices to bound the term $\left\langle \mathbf{w}_l, \mathbf{u} \right\rangle$, where $\|\mathbf{u}\|_2$ is a fixed vector. Hence, using the independence between $\mathbf{w}_l, \widetilde{\mathbf{x}}^{(l)}$, the bound for $\left\langle \mathbf{w}_l, \widetilde{\mathbf{x}}^{(l)} \right\rangle$ follows by first conditioning on $\widetilde{\mathbf{x}}^{(l)}$ and then using the bound for $\left\langle \mathbf{w}_l, \mathbf{u} \right\rangle$. Now for a fixed vector $\mathbf{u} \in \mathbb{R}^n$, $\|\mathbf{u}\|_2 = 1$, we can write

$$\begin{aligned}
\left\langle \mathbf{w}_l, \mathbf{u} \right\rangle = \left(\mathbf{W}^T\mathbf{u}\right)_l = \left\langle \mathbf{W}^T\mathbf{u}, \mathbf{e}_l \right\rangle &= \sum_{j=1}^{N}\left\langle \mathbf{z}_j\mathbf{z}_j^T\mathbf{u}, \mathbf{e}_l \right\rangle - \mathbb{E}\sum_{j=1}^{N}\left\langle \mathbf{z}_j\mathbf{z}_j^T\mathbf{u}, \mathbf{e}_l \right\rangle \\
&= \sum_{j=1}^{N}\mathbf{u}^T\mathbf{z}_j\mathbf{z}_j^T\mathbf{e}_l - \mathbb{E}\sum_{j=1}^{N}\mathbf{u}^T\mathbf{z}_j\mathbf{z}_j^T\mathbf{e}_l = \sum_{j=1}^{N}\mathbf{z}_j^T\mathbf{u}\mathbf{e}_l^T\mathbf{z}_j - \mathbb{E}\sum_{j=1}^{N}\mathbf{z}_j^T\mathbf{u}\mathbf{e}_l^T\mathbf{z}_j \\
&= \sum_{j=1}^{N}\left\langle \mathbf{z}_j, \mathbf{U}_l\mathbf{z}_j \right\rangle - \mathbb{E}\sum_{j=1}^{N}\left\langle \mathbf{z}_j, \mathbf{U}_l\mathbf{z}_j \right\rangle = \left\langle \mathbf{z}, \mathbf{U}\mathbf{z} \right\rangle - \mathbb{E}\left\langle \mathbf{z}, \mathbf{U}\mathbf{z} \right\rangle,
\end{aligned} \tag{37}$$

where

$$\mathbf{U}_l = \mathbf{u}\mathbf{e}_l^T \in \mathbb{R}^{n\times n}, \quad \mathbf{U} = \begin{bmatrix} \mathbf{U}_1 & & & & \\ & \mathbf{U}_2 & & & \\ & & \mathbf{U}_3 & & \\ & & & \ddots & \\ & & & & \mathbf{U}_N \end{bmatrix} \in \mathbb{R}^{nN\times nN}, \quad \mathbf{w} \in \mathbb{R}^{nN} = \begin{bmatrix} \mathbf{w}_1 \\ \mathbf{w}_2 \\ \mathbf{w}_3 \\ \vdots \\ \mathbf{w}_N \end{bmatrix} \sim \mathcal{N}(0, \mathbf{I}_{nN\times nN}). \tag{38}$$

Now, using Lemma 5 for $t > 0$ we have

$$\mathbb{P}\left\{|\left\langle \mathbf{w}_l, \mathbf{u} \right\rangle| > t\right\} = \mathbb{P}\left\{|\left\langle \mathbf{z}, \mathbf{U}\mathbf{z} \right\rangle - \mathbb{E}\left\langle \mathbf{z}, \mathbf{U}\mathbf{z} \right\rangle| > t\right\} \leq 2\exp\left\{-C\min\left\{\frac{t^2}{\|\mathbf{U}\|_F^2}, \frac{t}{\|\mathbf{U}\|_2}\right\}\right\} \tag{39}$$

$$= 2\exp\left\{-C\min\left\{\frac{t^2}{N}, t\right\}\right\}, \tag{40}$$

for some constant $C > 0$. Therefore, by taking $t = C'\sqrt{N\log N}$, where $C'C \geq 3$, we have with probability at least $1 - 2n^{-3}$,

$$|\left\langle \mathbf{w}_l, \mathbf{u} \right\rangle| \leq C'\sqrt{N\log N}. \tag{41}$$

Hence, using the union bound over $1 \leq l \leq n$, with probability at least $1 - O(n^{-2})$,

$$|\left\langle \mathbf{w}_l, \mathbf{u} \right\rangle| \leq C'\sqrt{N\log N}, \qquad \text{for } 1 \leq l \leq n. \tag{42}$$

Combining this with (36), (35) and since $\sqrt{a^2 + b^2} \le a + b$ for $a, b \ge 0$,

$$\|\mathbf{W}\widetilde{\mathbf{x}}\|_\infty \le \mathcal{O}\left(\sqrt{N \log N} + \left(\max_{1 \le l \le n} \left|\widetilde{\mathbf{x}}_l^{(l)}\right|\right) \lambda_{n,N}\right), \tag{43}$$

with high probability as $n \to \infty$. Now notice that for any $1 \le l \le n$,

$$\left(\mathbf{Y}^{(l)}\widetilde{\mathbf{x}}^{(l)}\right)_l = \lambda_1(\mathbf{Y}^{(l)})\widetilde{x}_l^{(l)} = \left\langle \mathbf{x}, \widetilde{\mathbf{x}}^{(l)} \right\rangle x_l + \sigma \left(\mathbf{W}^{(l)}\widetilde{\mathbf{x}}^{(l)}\right)_l = \left\langle \mathbf{x}, \widetilde{\mathbf{x}}^{(l)} \right\rangle x_l. \tag{44}$$

Therefore, using Weyl's inequality and (24)

$$\left|\widetilde{x}_l^{(l)}\right| = \frac{\left|\left\langle \mathbf{x}, \widetilde{\mathbf{x}}^{(l)} \right\rangle x_l\right|}{\lambda_1(\mathbf{Y}^{(l)})} \le \frac{M \left|\left\langle \mathbf{x}, \widetilde{\mathbf{x}}^{(l)} \right\rangle\right|}{1 - \sigma \left\|\mathbf{W}^{(l)}\right\|_2} \le \frac{M}{1 - c_0} \le 6M/5. \tag{45}$$

Hence, (43) results in

$$\|\mathbf{W}\widetilde{\mathbf{x}}\|_\infty \le \mathcal{O}\left(\sqrt{N \log N} + M\lambda_{n,N}\right), \tag{46}$$

with high probability as $n \to \infty$. Moreover, note that using the "sin $\Theta$" theorem, under (24)

$$\|\widetilde{\mathbf{x}} - \mathbf{x}\|_2 \le \mathcal{O}\left(\sigma \lambda_{n,N}\right). \tag{47}$$

In addition, note that we can choose $\widetilde{\mathbf{x}}$ such that $|\langle \mathbf{x}, \widetilde{\mathbf{x}} \rangle| = \langle \mathbf{x}, \widetilde{\mathbf{x}} \rangle$. Thus,

$$|\langle \mathbf{x}, \widetilde{\mathbf{x}} \rangle| = 1 - (1/2) \|\widetilde{\mathbf{x}} - \mathbf{x}\|_2^2 \ge 1 - \mathcal{O}(\sigma^2 \lambda_{n,N}^2), \qquad |\langle \mathbf{x}, \widetilde{\mathbf{x}} \rangle| \le 1. \tag{48}$$

Also we use Weyl's inequality $1 - \sigma \lambda_{n,N} \le \lambda_1(\mathbf{Y}) \le 1 + \sigma \lambda_{n,N}$. Finally, putting these and (43) in (29), under (24), we get

$$\|\widetilde{\mathbf{x}} - \mathbf{x}\|_\infty \le \left|\frac{|\langle \mathbf{x}, \widetilde{\mathbf{x}} \rangle|}{\lambda_1(\mathbf{Y})} - 1\right| M + \frac{\sigma \|\mathbf{W}\widetilde{\mathbf{x}}\|_\infty}{\lambda_1(\mathbf{Y})} \tag{49}$$

$$\lesssim ||\langle \mathbf{x}, \widetilde{\mathbf{x}} \rangle| - 1| M + |\lambda_1(\mathbf{Y}) - 1| M + \sigma \left(\sqrt{N \log N} + M\lambda_{n,N}\right)$$

$$\le \sigma \left(\sqrt{N \log N} + M \left(2\lambda_{n,N} + \lambda_{n,N}^2 \sigma\right)\right) \le \mathcal{O}\left(\sigma \left(\sqrt{N \log N} + M\lambda_{n,N}\right)\right), \tag{50}$$

with high probability as $n \to \infty$, and this completes the proof.

**Lemma 7.** *Let* $\mathbf{Y} = \mathbf{u}\mathbf{v}^T + \sigma\mathbf{W}$, *where* $\mathbf{u} \in \mathbb{R}^m$, $\mathbf{v} \in \mathbb{R}^n$, $\|\mathbf{v}\|_2 = \|\mathbf{u}\|_2 = 1$ *and* $\mathbf{W} \in \mathbb{R}^{m \times n}$ *where* $\mathbf{W}_{ij}$ *are i.i.d.* $\mathcal{N}(0,1)$ *gaussian random variables. Let* $\widetilde{\mathbf{u}} \in \mathbb{R}^m, \widetilde{\mathbf{v}} \in \mathbb{R}^n, \|\widetilde{\mathbf{v}}\|_2 = \|\widetilde{\mathbf{u}}\|_2 = 1$ *be the left and right singular vectors corresponding to the largest singularvalue of the matrix* $\mathbf{Y}$, *respectively. Let the operator norm of the matrix* $\mathbf{W}$ *be such that*

$$\|\mathbf{W}\|_2 \le \lambda_{m,n}, \tag{51}$$

*with probability at least* $1 - O((mn)^{-1})$. *Further, let*

$$\sigma \le \frac{c_0}{\lambda_{m,n}}, \tag{52}$$

*for some positive constant* $c_0 < 1/6$. *Letting,* $M_1 = \max \|\mathbf{u}\|_\infty$, $M_2 = \max \|\mathbf{v}\|_\infty$, *we have*

$$\|\widetilde{\mathbf{u}} - \mathbf{u}\|_\infty \le \Omega\left(\sigma \left(\sqrt{n \log n} + M_1 \lambda_{n,N}\right)\right), \tag{53}$$

$$\|\widetilde{\mathbf{v}} - \mathbf{v}\|_\infty \le \Omega\left(\sigma \left(\sqrt{m \log m} + M_2 \lambda_{n,N}\right)\right), \tag{54}$$

*with high probability as* $n \to \infty$.

**Proof.** This lemma can be proved similarly to Theorem 4 of [13] (the $\ell_\infty$ perturbation bound on eigenvectors) with slight modifications that we describe below.

If we let $\widetilde{\mathbf{u}}$ and $\widetilde{\mathbf{v}}$ be the top left and right eigenvectors of the matrix $\mathbf{Y} = \mathbf{u}\mathbf{v}^T + \sigma\mathbf{W}$ where $\mathbf{W}$ is a random matrix with i.i.d. $\mathcal{N}(0,1)$ entries, similar to the proof of results stated in [13] we can write

$$|\widetilde{u}_l - u_l| = \left|\frac{(\mathbf{Y}\widetilde{\mathbf{v}})_l}{\sigma_1(\mathbf{Y})} - u_l\right| = \left|\frac{\left(\left(\mathbf{u}\mathbf{v}^T + \sigma\mathbf{W}\right)\widetilde{\mathbf{v}}\right)_l}{\sigma_1(\mathbf{Y})} - u_l\right| \le \left|\left(\frac{\langle \widetilde{\mathbf{v}}, \mathbf{v} \rangle}{\sigma_1(\mathbf{Y})} - 1\right) u_l\right| + \left|\frac{(\mathbf{W}\widetilde{\mathbf{v}})_l}{\sigma_1(\mathbf{Y})}\right| \sigma \tag{55}$$

$$\le \left|\left(\frac{\langle \widetilde{\mathbf{v}}, \mathbf{v} \rangle}{\sigma_1(\mathbf{Y})} - 1\right)\right| M' + \frac{\sigma |(\mathbf{W}\widetilde{\mathbf{v}})_l|}{\sigma_1(\mathbf{Y})}, \tag{56}$$

where $M_1 = \max_l |u_l|$, for $1 \le l \le n$. Note that this bound is exactly the same as the bound stated in (28) and in [13]. Therefore, by defining $\mathbf{W}^{(l)} \in \mathbb{R}^{m \times n}$ as

$$W_{i,j}^{(l)} \triangleq W_{i,j}, \quad \text{for} \quad i \ne l, \tag{57}$$

$$W_{i,j}^{(l)} \triangleq 0, \quad \text{if} \quad i = l. \tag{58}$$

we can follow exactly the same steps in [13] to prove the $\ell_\infty$ perturbation bound on $\|\widetilde{\mathbf{u}} - \mathbf{u}\|_\infty$. The only part that needs a slight change is the $\ell_2$ perturbation bound on the singular vectors (similar to bound to the bound we used in (31)). Here instead of the Davis-Kahan "sin $\Theta$" Theorem, we can use the Wedin's Theorem [14] to have

$$\left\|\widetilde{\mathbf{v}} - \widetilde{\mathbf{v}}^{(l)}\right\|_2 \le \frac{\sigma\sqrt{2}\max\left\{\left\|\Delta\mathbf{W}^{(l)}\widetilde{\mathbf{v}}^{(l)}\right\|_2, \left\|\Delta\mathbf{W}^{(l)^T}\widetilde{\mathbf{u}}^{(l)}\right\|_2\right\}}{\delta\left(\mathbf{Y}^{(l)}\right) - \sigma\left\|\Delta\mathbf{W}^{(l)}\right\|_2}. \tag{59}$$

Using the definition of $\Delta\mathbf{W}^{(l)} = \mathbf{W} - \mathbf{W}^{(l)}$, we have

$$\left\|\Delta\mathbf{W}^{(l)}\widetilde{\mathbf{v}}^{(l)}\right\|_2 = \left|\left\langle\mathbf{w}_l, \widetilde{\mathbf{v}}^{(l)}\right\rangle\right|, \tag{60}$$

$$\left\|\Delta\mathbf{W}^{(l)^T}\widetilde{\mathbf{u}}^{(l)}\right\|_2 \le \|\mathbf{w}_l\|_2 \left\|\widetilde{\mathbf{u}}^{(l)}\right\|_\infty, \tag{61}$$

where $\mathbf{w}_l$ is the $l$-th row of $\mathbf{W}$. Using these inequalities, bounding the term $\left|\left\langle\mathbf{w}_l, \widetilde{\mathbf{v}}^{(l)}\right\rangle\right|$, using the independence between $\mathbf{w}_l$ and $\widetilde{\mathbf{v}}^{(l)}$ and Gaussian concentration (Bernstein Inequality) and bounding $\left\|\widetilde{\mathbf{u}}^{(l)}\right\|_\infty$ (as in (45)) will give us the singular value version of the $\ell_\infty$ perturbation bound in the following lemma. Using the same argument for $\mathbf{Y}^T$ will give a similar bound on $\|\widetilde{\mathbf{v}} - \mathbf{v}\|_\infty$.

**Lemma 8.** *Let $\mathcal{P}_0,...,\mathcal{P}_M$ be probability measures on the same probability space where $M \ge 2$. If for some $0 < \alpha < 1$, we have*

$$\frac{1}{M+1}\sum_{i=0}^{M} D_{KL}(\mathcal{P}_i\|\bar{\mathcal{P}}) \le \alpha\log(M) \tag{62}$$

*where*

$$\bar{\mathcal{P}} = \frac{1}{M+1}\sum_{i=0}^{M}\mathcal{P}_i \tag{63}$$

*Then,*

$$p_{e,M} \ge \frac{\log(M+1) - \log(2)}{\log(M)} - \alpha \tag{64}$$

*where $p_{e,M}$ is the minimax error for the multiple testing problem.*

**Proof.** See reference [15].

**Lemma 9.** *Let $\mathcal{P}_i$ be a multivariate Gaussian distribution with mean $\mu_i$ and covariance $\mathbf{\Gamma}_i$, for $i = 1, 2$. Then*

$$D(\mathcal{P}_1\|\mathcal{P}_2) = \frac{1}{2}\left(Tr\left(\mathbf{\Gamma}_2^{-1}\mathbf{\Gamma}_1\right) + (\mu_1 - \mu_2)^t\mathbf{\Gamma}_2^{-1}(\mu_1 - \mu_2) + \ln\left(\frac{det(\mathbf{\Gamma}_2)}{det(\mathbf{\Gamma}_1)}\right)\right) \tag{65}$$

**Lemma 10.** *Let $\mathcal{Z}$ be a tensor whose entries are i.i.d. normal. We have*

$$\mathbb{E}\left[\exp\left(<\mathcal{A}, \mathcal{Z}>\right)\right] = \exp\left(\frac{\|\mathcal{A}\|_F^2}{2}\right). \tag{66}$$

**Lemma 11.** *Let $\mathbf{v} = (v_1, ..., v_m)$ be a vector distributed uniformly over the unit sphere. We have*

$$\mathbb{E}[\exp(\alpha v_1)] = c\exp\left(\frac{\alpha^2}{2m}\right). \tag{67}$$

*where $c$ is a constant and $\alpha$ can grow with $m$.*

**Proof.** We have $v_1 \stackrel{\mathrm{d}}{=} \sqrt{\frac{x_1^2}{x_1^2 + S_{m-1}}}$ where $x_1$ is normal and $S_{m-1}$ has a $\chi$-squared distribution with $m-1$ degrees of freedom [4].

We have

$$\mathbb{E}[\exp(\alpha v_1)] = \int_1^{\exp(\alpha)} \mathbb{P}(\exp(\alpha v_1) \geq y) dy \tag{68}$$

$$= \int_1^{\exp(\alpha)} \mathbb{P}(v_1 \geq \frac{\log(y)}{\alpha}) dy.$$

On the other hand, we have

$$\mathbb{P}(v_1 \geq \frac{\log(y)}{\alpha}) = \mathbb{P}(\frac{S_{m-1}}{x_1^2} \leq \frac{\alpha^2}{\log(y)^2}). \tag{69}$$

Using Lemma 2, we have

$$\mathbb{P}\left(S_{m-1} \leq m - 1 - 2\sqrt{(m-1)t}\right) \leq \exp(-t). \tag{70}$$

Similarly we have

$$\mathbb{P}\left(x_1^2 \geq 1 + 2\sqrt{t} + 2t\right) \leq \exp(-t). \tag{71}$$

Combining (70) and (71), we have

$$\mathbb{P}\left(\frac{S_{m-1}}{x_1^2} \leq c_1 \frac{m - \sqrt{mt}}{t}\right) < \exp(-t), \tag{72}$$

where $c_1$ is a constant. Choosing

$$t = \frac{4m}{\left(1 + \sqrt{1 + \frac{4\alpha^2}{c_1 \log(y)^2}}\right)^2} \tag{73}$$

we have

$$\mathbb{P}\left(\frac{S_{m-1}}{x_1^2} \leq \frac{\alpha^2}{\log(y)^2}\right) \leq \exp\left(-c_2 \frac{m \log(y)^2}{\alpha^2}\right) \tag{74}$$

where $c_2$ is a constant. Moreover, we have

$$\int_1^{\exp(\alpha)} \exp\left(-c_2 \frac{m \log(y)^2}{\alpha^2}\right) dy \leq c_3 \exp\left(\frac{\alpha^2}{m}\right). \tag{75}$$

Combining (68) and (75) completes the proof.

### 4.2 Proof of Theorem MT-1

First, we prove Theorem MT-1 for the noise model I where $\sigma_z^2 = 1$. Without loss of generality we assume $J_1 = \{1, 2, ..., k\}$ and $J_2 = \{1, 2, ..., k\}$. Recall that $\mathbf{T}_{(j,1)} \in \mathbb{R}^{m \times n}$ is the $j$-th horizontal matrix slice of the tensor $\mathcal{T}$. We have $\mathbf{T}_{(j,1)} = \mathbf{X}_{(j,1)} + \mathbf{Z}_{(j,1)}$ where $\mathbf{X}_{(j,1)}$ and $\mathbf{Z}_{(j,1)}$ are the $j$-th horizontal matrix slices of signal and noise tensors $\mathcal{X}$ and $\mathcal{Z}$.

For $j \in [k]$, we have

$$\mathbf{X}_{(j,1)} = \sigma_1 \mathbf{u}_1(j) \left(\mathbf{1}_{1 \times n} \otimes \mathbf{v}_1\right) \mathrm{Diag}\left(\mathbf{w}_1(1), ..., \mathbf{w}_1(k), 0, ..., 0\right). \tag{76}$$

Thus for $j \in [k]$,

$$\mathbf{X}_{(j,1)}^t \mathbf{X}_{(j,1)} = \sigma_1^2 (\mathbf{u}_1(j))^2 \mathbf{w}_1 (\mathbf{w}_1)^t. \tag{77}$$

Summing this over $j$ and using the fact that $\|\mathbf{u}_1\| = 1$, we have

$$\sum_{j=1}^n \mathbf{X}_{(j,1)}^t \mathbf{X}_{(j,1)} = \sigma_1^2 \mathbf{w}_1 (\mathbf{w}_1)^t. \tag{78}$$

Thus the largest eigenvalue of the matrix $\sum_{j=1}^{n} \mathbf{X}_{(j,1)}^{t} \mathbf{X}_{(j,1)}$ is $\sigma_1^2$ which corresponds to the eigenvector $\mathbf{w}_1$. Note that entries of this eigenvector is all zero outside of the bicluster index set $J_2$.

Next we bound the operator norm of noise terms. For $j \in [k]$, we have

$$\mathbf{T}_{(j,1)}^{t} \mathbf{T}_{(j,1)}^{t} = \mathbf{X}_{(j,1)}^{t} \mathbf{X}_{(j,1)} + \mathbf{Z}_{(j,1)}^{t} \mathbf{Z}_{(j,1)} + \mathbf{X}_{(j,1)}^{t} \mathbf{Z}_{(j,1)} + \mathbf{Z}_{(j,1)}^{t} \mathbf{X}_{(j,1)}. \tag{79}$$

For $j > k$, we have

$$\mathbf{T}_{(j,1)}^{t} \mathbf{T}_{(j,1)}^{t} = \mathbf{Z}_{(j,1)}^{t} \mathbf{Z}_{(j,1)}. \tag{80}$$

Summing these terms over $j \in [n]$ we have

$$\sum_{j=1}^{n} \mathbf{T}_{(j,1)}^{t} \mathbf{T}_{(j,1)}^{t} = \sum_{j=1}^{n} \mathbf{X}_{(j,1)}^{t} \mathbf{X}_{(j,1)} + \underbrace{\sum_{j=1}^{n} \mathbf{Z}_{(j,1)}^{t} \mathbf{Z}_{(j,1)}}_{\text{noise term I}} + \underbrace{\sum_{j=1}^{k} \mathbf{X}_{(j,1)}^{t} \mathbf{Z}_{(j,1)} + \mathbf{Z}_{(j,1)}^{t} \mathbf{X}_{(j,1)}}_{\text{noise term II}} \tag{81}$$

Using Lemma 4, with high probability, we have

$$\left\| \sum_{j=1}^{n} \mathbf{Z}_{(j,1)}^{t} \mathbf{Z}_{(j,1)} - nm\mathbf{I} \right\| < c\sqrt{n} \max(n,m) \log(n) \tag{82}$$

where $c > 0$ is a universal constant. Also according to the argument explained in the last paragraph of Section 3.1, we also have

$$\left\| \sum_{j=1}^{n} \mathbf{Z}_{(j,1)}^{t} \mathbf{Z}_{(j,1)} - nm\mathbf{I} \right\| < cnm \log(n) \tag{83}$$

Let

$$\lambda_{z,1} \triangleq \min(\sqrt{n} \max(n,m), nm) \log(n). \tag{84}$$

Thus, the operator norm of the noise term I subtracted from its mean is bounded by $\lambda_{z,1}$. Note that since the mean of the noise term I is a scaled identity matrix, subtracting this term does not change the eigenvector structure.

Next, we bound the operator norm of the noise term II in (81). We have

$$\mathbf{X}_{(j,1)}^{t} \mathbf{Z}_{(j,1)} = \sigma_1 \mathbf{u}_1(j) \mathbf{w}_1 \mathbf{z}_j^{t} \tag{85}$$

where $\mathbf{z}_j$ is a vector of length $n$ whose entries have i.i.d. normal distributions. Since $|\mathbf{u}_1(j)| \leq 1/\sqrt{k}$, using Lemma 2, as $n \to \infty$, with high probability,

$$\left\| \mathbf{X}_{(j,1)}^{t} \mathbf{Z}_{(j,1)} \right\| \leq \frac{\sigma_1 \sqrt{n + \sqrt{n \log(n)}}}{\sqrt{k}}. \tag{86}$$

As $n \to \infty$, using Lemma 3 for matrices $\mathbf{X}_{(j,1)}^{t} \mathbf{Z}_{(j,1)}$ for $1 \leq j \leq k$, with high probability, we have

$$\left\| \sum_{j=1}^{k} \mathbf{X}_{(j,1)}^{t} \mathbf{Z}_{(j,1)} \right\| \leq c\lambda_{z,2} \tag{87}$$

where

$$\lambda_{z,2} \triangleq \sigma_1 \sqrt{n} \log(n) \tag{88}$$

According to (78), the operator norm of the folded signal tensor is $\sigma_1^2$. According to (84) and (88), the operator norm of the noise is bounded by $\max(\lambda_{z,1}, \lambda_{z,2})$. If $\lambda_{z,1} \gg \lambda_{z,2}$, then using Lemma 6, to have vanishing error probability it suffices to have

$$\frac{\sqrt{n \log(n)} + \lambda_{z,1}/\sqrt{k}}{\sigma_1^2} \leq c \frac{1}{\sqrt{k}}, \tag{89}$$

which leads to the condition $\sigma_1^2 = \Omega(\min(\sqrt{n}\max(n,m), nm)\log(n))$. If $\lambda_{z,2} \gg \lambda_{z,1}$, using an $l_\infty$ Davis-Kahan bound similarly to Lemma 6, we need to have $\sigma_1^2 = \Omega(n\log(n))$ which is always dominated by the previous case (later in this section we show that the argument of Lemma 6 holds for the noise term II as well.).

For the noise model II, the operator norm of the folded signal is equal to $\sigma_1^2$. Since $\Theta(n-k) = n$, the operator norm of noise terms can be bounded similarly. The rest of the proof is similar to the case of noise model I.

Next we show a similar result to the one of Lemma 6 holds for the noise term II as well. To prove this, note that we can write

$$\sum_{j=1}^{k} \mathbf{X}_{(j,1)}^T \mathbf{Z}_{(j,1)} = \sigma_1 \left( \mathbf{w}_1 \sum_{j=1}^{k} \mathbf{u}_1(j) \mathbf{z}_j^T + \sum_{j=1}^{k} \mathbf{u}_1(j) \mathbf{z}_j \mathbf{w}_1^T \right) = \sigma_1 \left( \mathbf{w}_1 \widetilde{\mathbf{z}}_j^T + \widetilde{\mathbf{z}}_j \mathbf{w}_1^T \right), \quad (90)$$

where

$$\widetilde{\mathbf{z}}_j = \sum_{j=1}^{k} \mathbf{u}_1(j) \mathbf{z}_j \quad (91)$$

is a gaussian random vector with i.i.d. $\mathcal{N}(0,1)$ entries. In the proof of Lemma 6, except the last part which bounds the term $\langle \mathbf{w}_l, \mathbf{u} \rangle$, all steps will go through similarly after replacing the noise matrix $\mathbf{W}$ in the lemma with $\sum_{j=1}^{k} \mathbf{X}_{(j,1)}^T \mathbf{Z}_{(j,1)}$. For bounding the term $\langle \mathbf{w}_l, \mathbf{u} \rangle$ in this case, for a fixed vector $\mathbf{u} \in \mathbb{R}^n, \|\mathbf{u}\|_2 = 1$, we can write

$$\left( \sum_{j=1}^{k} \mathbf{X}_{(j,1)}^T \mathbf{Z}_{(j,1)} \mathbf{u} \right)_l = \sigma_1 \left( (\mathbf{w}_1)_l \langle \widetilde{\mathbf{z}}_j, \mathbf{u} \rangle + (\widetilde{\mathbf{z}}_j)_l \langle \mathbf{w}_1, \mathbf{u} \rangle \right), \quad (92)$$

since $\|\mathbf{w}_1\|_2 = \|\mathbf{u}\|_2 = 1$, using Cauchy-Schwarz inequality, $|\langle \mathbf{w}_1, \mathbf{u} \rangle| \leq 1$. Thus,

$$\left( \sum_{j=1}^{k} \mathbf{X}_{(j,1)}^T \mathbf{Z}_{(j,1)} \mathbf{u} \right)_l \leq \sigma_1 \left( (\mathbf{w}_1)_l \langle \widetilde{\mathbf{z}}_j, \mathbf{u} \rangle + \left| (\widetilde{\mathbf{z}}_j)_l \right| \right) = \sigma_1 \max \left\{ \langle \widetilde{\mathbf{z}}_j, (\mathbf{w}_1)_l \mathbf{u} + \mathbf{e}_l \rangle, \langle \widetilde{\mathbf{z}}_j, (\mathbf{w}_1)_l \mathbf{u} - \mathbf{e}_l \rangle \right\}$$
$$(93)$$

Using $|(\mathbf{w}_1)_l| \leq 1$, we have

$$\left( \sum_{j=1}^{k} \mathbf{X}_{(j,1)}^T \mathbf{Z}_{(j,1)} \mathbf{u} \right)_l \leq \sigma_1 |\langle \widetilde{\mathbf{z}}_j, \mathbf{u} + \mathbf{e}_l \rangle|. \quad (94)$$

Now, using Lemma 2, for $t > 0$ we have,

$$\mathbb{P} \left\{ |\langle \widetilde{\mathbf{z}}_j, \sigma_1(\mathbf{u} + \mathbf{e}_l) \rangle| > t \right\} \leq e \exp \left( \frac{-Ct^2}{\sigma_1^2 \|\mathbf{u} + \mathbf{e}_l\|_2^2} \right) \leq e \exp \left( \frac{-Ct^2}{4\sigma_1^2} \right), \quad (95)$$

for some constant $C > 0$. Hence, by taking $t = C'\sigma_1\sqrt{n\log n}$, where $C'C > 12$, with probability at least $1 - en^{-3}$ we have

$$\left( \sum_{j=1}^{k} \mathbf{X}_{(j,1)}^T \mathbf{Z}_{(j,1)} \mathbf{u} \right)_l \leq C'\sigma_1\sqrt{n\log n}. \quad (96)$$

Using union bound over $1 \leq l \leq n$, with probability at least $1 - O(n^{-2})$ we have

$$\left\| \left( \sum_{j=1}^{k} \mathbf{X}_{(j,1)}^T \mathbf{Z}_{(j,1)} \mathbf{u} \right) \right\|_\infty \leq C'\sigma_1\sqrt{n\log n}. \quad (97)$$

Therefore, if we denote the top eigenvector after adding the noise term II by $\widetilde{\mathbf{x}}$ and take $\lambda'_{n,N}$ such that

$$\left\| \sum_{j=1}^{k} \mathbf{X}_{(j,1)}^{T} \mathbf{Z}_{(j,1)} \right\|_{2} \leq \lambda'_{n,N}, \tag{98}$$

the same argument used to prove the $\ell_{\infty}$ perturbation bound in Lemma 6, can be used here to show that

$$\|\widetilde{\mathbf{x}} - \mathbf{x}\|_{\infty} \leq \Omega\left(\sigma\left(\sqrt{n \log n} + M\lambda'_{n,N}\right)\right), \tag{99}$$

with high probability as $n \to \infty$.

### 4.3 Proof of Theorem MT-2

First we consider the noise model I where $\sigma_z^2 = 1$. Since $\|\mathbf{u}_1 \otimes \mathbf{w}_1\| = 1$, we have

$$\|\mathbf{X}_{unfolded}\| = \sigma_1. \tag{100}$$

Moreover, as $n \to \infty$, since Since $\|\mathbf{Z}_{unfolded}\|$ has a sub-Gaussian tail distribution, with high probability, we have

$$\|\mathbf{Z}_{unfolded}\| = \mathcal{O}\left(\max(n, \sqrt{m}) \log(n)\right). \tag{101}$$

Using (100), (101), with Lemma 7 completes the proof for the case of $\sigma_z^2 = 1$. The bounds for the noise model II are similar to the ones of $\sigma_z^2 = 1$ since $\Theta(n - k) = n$.

### 4.4 Proof of Theorem MT-3

First we consider the noise model I where $\sigma_z^2 = 1$. If $(j_1, j_2) \in J_1 \times J_2$, we have

$$\|\mathcal{T}(j_1, j_2, :)\|^2 = \|\mathcal{X}(j_1, j_2, :) + \mathcal{Z}(j_1, j_2, :)\|^2 \tag{102}$$
$$= \|\mathcal{X}(j_1, j_2, :)\|^2 + \|\mathcal{Z}(j_1, j_2, :)\|^2 + 2\mathcal{X}(j_1, j_2, :)^t \mathcal{Z}(j_1, j_2, :)$$
$$= \sigma_1^2 \mathbf{u}_1(j_1)^2 \mathbf{w}_1(j_2)^2 + \|\mathcal{Z}(j_1, j_2, :)\|^2 + 2\sigma_1 |\mathbf{u}_1(j_1)\mathbf{w}_1(j_2)| \mathbf{v}_1^t \mathcal{Z}(j_1, j_2, :)$$

Note that since $\|\mathbf{v}_1\| = 1$, $\mathbf{v}_1^t \mathcal{Z}(j_1, j_2, :)$ has a standard normal distribution. Thus, using Lemma 1, the term $2\sigma_1 |\mathbf{u}_1(j_1)\mathbf{w}_1(j_2)| \mathbf{v}_1^t \mathcal{Z}(j_1, j_2, :)$ can be ignored compared to the term $\sigma_1^2 \mathbf{u}_1(j_1)^2 \mathbf{w}_1(j_2)^2$. Moreover, we have

$$\sigma_1^2 \mathbf{u}_1(j_1)^2 \mathbf{w}_1(j_2)^2 \geq \sigma_1^2 \Delta^4. \tag{103}$$

Since the noise variance is one, $\|\mathcal{Z}(j_1, j_2, :)\|^2$ has a $\chi$-squared distribution with $m$ degrees of freedom. Thus, for every $(j_1, j_2) \in J_1 \times J_2$, using Lemma 2, if $\sigma_1^2 = \tilde{\Omega}(\sqrt{m}/\Delta^4)$, we have

$$\mathbb{P}\left[\|\mathcal{T}(j_1, j_2, :)\|^2 - m \geq \frac{\sigma_1^2 \Delta^4}{2}\right] > 1 - n^{-c} \tag{104}$$

where $c > 0$ is a universal constant.

If $(j_1, j_2) \notin J_1 \times J_2$, we have

$$\|\mathcal{T}(j_1, j_2, :)\|^2 = \|\mathcal{Z}(j_1, j_2, :)\|^2, \tag{105}$$

where $\|\mathcal{Z}(j_1, j_2, :)\|^2$ has a $\chi$-squared distribution with $m$ degrees of freedom. Thus, using Lemma 2 and the union bound, if $\sigma_1^2 = \tilde{\Omega}(\sqrt{m}/\Delta^4)$, we have

$$\mathbb{P}\left[\max_{(j_1, j_2) \notin J_1 \times J_2} \|\mathcal{T}(j_1, j_2, :)\|^2 - m \geq \frac{\sigma_1^2 \Delta^4}{2}\right] \leq n^{-c} \tag{106}$$

where $c > 0$ is a universal constant. This complete the proof for the noise model I.

For the case of noise model II, if $(j_1, j_2) \in J_1 \times J_2$ in (102), $\|\mathcal{Z}(j_1, j_2, :)\|^2/\sigma_z^2$ has a $\chi$-squared distribution with $m$ degrees of freedom. Similarly, $\mathcal{X}(j_1, j_2, :)^t \mathcal{Z}(j_1, j_2, :)/\sigma_z$ has a Gaussian distribution with mean zero and variance $\|\mathcal{X}(j_1, j_2, :)\|^2$. If $(j_1, j_2) \notin J_1 \times J_2$ in (105), $\|\mathcal{T}(j_1, j_2, :)\|^2$ has a $\chi$-squared distribution with $m$ degrees of freedom. The rest of the proof is similar to the case of noise model I.

## 4.5 Proof of Theorem MT-4

First we consider the noise model I where $\sigma_z^2 = 1$. If $j_1 \in J_1$, using (102), we have

$$d_{j_1} = \sum_{j_2=1}^{n} \|\mathcal{T}(j_1, j_2, :)\|^2 \tag{107}$$

$$= \sigma_1^2 \mathbf{u}_1(j_1)^2 + \sum_{j_2=1}^{n} \|\mathcal{Z}(j_1, j_2, :)\|^2 + 2\sigma_1 |\mathbf{u}_1(j_1)| \mathbf{v}_1^t \sum_{j_2=1}^{k} |\mathbf{w}_1(j_2)| \mathcal{Z}(j_1, j_2, :).$$

Similarly to (102), using Lemma 1, the term $2\sigma_1 |\mathbf{u}_1(j_1)| \mathbf{v}_1^t \sum_{j_2=1}^{k} |\mathbf{w}_1(j_2)| \mathcal{Z}(j_1, j_2, :)$ can be ignored against $\sigma_1^2 \mathbf{u}_1(j_1)^2$. Moreover, we have

$$\sigma_1^2 \mathbf{u}_1(j_1)^2 \geq \sigma_1^2 \Delta^2 \tag{108}$$

Since the noise variance is one, $\sum_{j_2=1}^{n} \|\mathcal{Z}(j_1, j_2, :)\|^2$ has a $\chi$-squared distribution with $mn$ degrees of freedom. Thus, for every $(j_1, j_2) \in J_1 \times J_2$, using Lemma 2, if $\sigma_1^2 = \tilde{\Omega}(\sqrt{nm}/\Delta^2)$, we have

$$\mathbb{P}\left[d_{j_1} - mn \geq \frac{\sigma_1^2 \Delta^2}{2}\right] > 1 - n^{-c} \tag{109}$$

where $c > 0$ is a universal constant.

If $j_1 \notin J_1$, $d_{j_1}$ has a $\chi$-squared distribution with $nm$ degrees of freedom. Thus, using Lemma 2 and the union bound, if $\sigma_1^2 = \tilde{\Omega}(\sqrt{nm}/\Delta^2)$, we have

$$\mathbb{P}\left[\max_{j_1 \notin J_1} d_{j_1} - mn \geq \frac{\sigma_1^2 \Delta^2}{2}\right] \leq n^{-c} \tag{110}$$

where $c > 0$ is a universal constant. This completes the proof for the noise model I. The proof for the noise model II follows from similar steps.

## 4.6 Proof of Theorem MT-5

Let $\hat{C} = \hat{J}_1 \times \hat{J}_2$. Let $\bar{C}$ be remaining tuple indices. First we consider the noise model I where $\sigma_z^2 = 1$. Thus MT-(4) simplifies to

$$\mathbb{P}\left[(\hat{J}_1, \hat{J}_2)|\mathcal{T}\right] \propto \mathbf{v}_1^t \sum_{(j_1, j_2) \in \hat{C}} \mathcal{T}(j_1, j_2, :). \tag{111}$$

Suppose $\mathcal{T}$ is generated by $(J_1, J_2)$. Let $a = |J_1 \cap \hat{J}_1|$ and $b = |J_2 \cap \hat{J}_2|$. If $(j_1, j_2) \in C \cap \hat{C}$, we have $\mathcal{T}(j_1, j_2, :) = \sigma_1/k \mathbf{v}_1 + \mathbf{z}_{j_1, j_2}$ where entries of $\mathbf{z}_{j_1, j_2}$ have normal distributions. If $(j_1, j_2) \in \hat{C} - C$, we have $\mathcal{T}(j_1, j_2, :) = \mathbf{z}_{j_1, j_2}$ where entries of $\mathbf{z}_{j_1, j_2}$ have normal distributions. Thus,

$$\mathbf{v}_1^t \sum_{(j_1, j_2) \in C} \mathcal{T}(j_1, j_2, :) = \frac{\sigma_1 ab}{k} + Z \tag{112}$$

where $Z$ has a standard normal distribution. Using the union bound and Lemma (1), we have

$$\mathbb{P}\left[\max_{\hat{C}} \left|\mathbf{v}_1^t \sum_{(j_1, j_2) \in \hat{C}} \mathcal{T}(j_1, j_2, :) > t\right|\right] < \exp\left(-ck \log(ne/k)\right), \tag{113}$$

where $c > 0$ is a universal constant. Thus, if $\sigma_1 k > \Omega(\sqrt{k \log(ne/k)})$, the probability of error goes to zero.

For the noise model II, if $\sigma_1^2 > mk^2$, the likelihood score of every $\hat{C} \neq C$ is $-\infty$ while the likelihood score of $C$ is finite. Thus, Theorem MT-5 holds in this case. Next we assume $\sigma_1^2 < mk^2$. In this regime the likelihood score MT-(4) simplifies to

$$\mathbb{P}\left[(\hat{J}_1, \hat{J}_2)|\mathcal{T}\right] \propto \mathbf{v}_1^t \sum_{(j_1, j_2) \in \hat{C}} \mathcal{T}(j_1, j_2, :) - \frac{\sigma_1}{2mk} \sum_{(j_1, j_2) \in \hat{C}} \|\mathcal{T}(j_1, j_2, :)\|^2. \tag{114}$$

Note that, under the noise model II, the expected value of $\sum_{(j_1,j_2)\in\hat{C}}\|\mathcal{T}(j_1,j_2,:)\|^2$ is the same for every $\hat{C}$. Thus, using Lemma 2, if

$$\sigma_1 k = \Omega\left(\frac{\sigma_1}{mk}\sqrt{mk^2\log(mk)}\sqrt{k\log(n/k)}\right) \tag{115}$$

the effect of the second term is negligible as $n \to \infty$. This holds if $mk \gg \log(n/k)$. Thus, this simplifies the problem to the case of noise model I. This completes the proof.

### 4.7 Proof of Theorem MT-6

First we consider the noise model I where $\sigma_z^2 = 1$. We have

$$|J_{all}| = \binom{n}{k}^2 \leq (\frac{ne}{k})^{2k}. \tag{116}$$

Thus, $\log(|J_{all}|) \leq 2k\log(ne/k)$.

Let $\mathcal{P}_i$ be the probability measure induced by the model $J^{(i)} \in J_{all}$. Let

$$\bar{\mathcal{P}} = \frac{1}{|J_{all}|}\sum_{i=1}^{|J_{all}|}\mathcal{P}_i. \tag{117}$$

Thus, for every $1 \leq i \leq |J_{all}|$, we have

$$D_{KL}(\mathcal{P}_i\|\bar{\mathcal{P}}) \leq \frac{1}{|J_{all}|}\sum_{j=1}^{|J_{all}|}D_{KL}(\mathcal{P}_i\|\mathcal{P}_j) \leq \max_j \ D_{KL}(\mathcal{P}_i\|\mathcal{P}_j) \tag{118}$$

where the first inequality comes from the convexity of the KL divergence.

Consider two tensor biclustering models $J^{(i)}, J^{(j)} \in J_{all}$ where their bicluster indices are non-overlapping. This is possible since $k < n/2$. Using Lemma 9, for such $\mathcal{P}_i$ and $\mathcal{P}_j$ we have $D_{KL}(\mathcal{P}_i\|\mathcal{P}_j) = \sigma_1^2$. If bicluster indices of tensor biclustering models $J^{(i)}$ and $J^{(j)}$ overlap with each other, the KL divergence between their induced probability measures is smaller than $\sigma_1^2$. Thus,

$$\max_{i,j} \ D_{KL}(\mathcal{P}_i\|\mathcal{P}_j) = \sigma_1^2 \tag{119}$$

Using Lemma 8, if $\sigma_1^2 < \alpha\log(|J_{all}|)$, the minimax error is lower bounded by $1 - \alpha - \log(2)/\log(|J_{all}|)$. Using (116) completes the proof for the case of having noise model I.

Now consider the case of noise model II. If $\sigma_1^2 > mk^2$, a simple algorithm based on thresholding individual trajectory lengths can solve the tensor biclustering problem with vanishing error probability (Theorem MT-3). Thus, without loss of generality, we assume $\sigma_1^2 < mk^2$. Using a similar argument to the one of noise model I, one can show that

$$\max_{i,j} \ D_{KL}(\mathcal{P}_i\|\mathcal{P}_j) = \frac{1}{1 - \sigma_1^2/mk^2}\sigma_1^2 \tag{120}$$

Then, using lemma 8, if

$$\sigma_1^2 < \mathcal{O}(\frac{k\log(n/k)}{1 + \log(n/k)/mk}) \tag{121}$$

the minimax error is lower bounded by $1 - \alpha - \log(2)/\log(|J_{all}|)$. This completes the proof.

### 4.8 Proof of Theorem MT-7

Recall the ML optimization MT-(6). Suppose $\mathcal{T}$ is generated by $(J_1, J_2)$. Let $a = |\hat{J}_1 \cap J_1|$ and $b = |\hat{J}_2 \cap J_2|$. Let $\hat{C} = (\hat{J}_1, \hat{J}_2)$ and $C = (J_1, J_2)$. If $(j_1, j_2) \in \hat{C} \cap C$, we have $\mathcal{T}(j_1, j_2, :) = \sigma_1/k\mathbf{v}_1 + \mathbf{z}_{j_1,j_2}$ where entries of $\mathbf{z}_{j_1,j_2}$ have normal distributions. If $(j_1, j_2) \in \hat{C} - C$, we have $\mathcal{T}(j_1, j_2, :) = \mathbf{z}_{j_1,j_2}$ where entries of $\mathbf{z}_{j_1,j_2}$ have normal distributions. Thus,

$$\sum_{(j_1,j_2)\in\hat{C}}\mathcal{T}(j_1,j_2,:) = \frac{\sigma_1 ab}{k}\mathbf{v}_1 + k\mathbf{z} \tag{122}$$

where $\mathbf{z}$ is a vector of length $m$ with i.i.d. normal distributions. Thus, we have

$$\| \sum_{(j_1,j_2)\in\hat{C}} \mathcal{T}(j_1,j_2,:)\|^2 = (\frac{\sigma_1 ab}{k})^2 + k^2 S_m + 2\sigma_1 abZ, \tag{123}$$

where $S_m$ has a $\chi$-squared distribution with $m$ degrees of freedom and $Z$ is normal. Let $t = k^2\sigma_1^2/2$. Using Lemma 2, we have

$$\mathbb{P}\left[|k^2 S_m - k^2 m| > t\right] < \exp\left(-\min(\frac{t^2}{mk^4}, \frac{t}{k^2})\right) \tag{124}$$

$$< \exp\left(-\min(\frac{\sigma_1^4}{4m}, \frac{\sigma_1^2}{2})\right)$$

$$< \exp\left(-k\log(n/k)\right),$$

if $\sigma_1$ satisfies conditions of the theorem. A similar argument can be stated for the cross noise terms $\sigma_1 abZ$. Using a union bound over $\binom{n}{k}^2$ choices for $\hat{J}$ completes the proof.

## 4.9  Proof of Theorem MT-8

Let $\mathcal{A}_1 \triangleq \mathbf{u}_1 \otimes \mathbf{w}_1 \otimes \mathbf{v}_1$. Under the model described in Section 8, we have

$$\mathbb{P}_{\sigma_1}(\mathcal{X}) = \frac{1}{\binom{n}{k}^2} \sum_J \int \exp\left(-\|\mathcal{X} - \mathcal{A}_1\|_F^2/2\right) \mu(d\mathbf{v}_1) \tag{125}$$

where $\mu(.)$ is the uniform measure on the unit sphere. We also have

$$\mathbb{P}_0(\mathcal{X}) = \exp\left(\|\mathcal{X}\|_F^2/2\right). \tag{126}$$

Let $\Lambda$ be the Radon-Nikodym derivative of $\mathbb{P}_{\sigma_1}$ with respect to $\mathbb{P}_0$. Thus, we have

$$\Lambda = \frac{d\mathbb{P}_{\sigma_1}}{d\mathbb{P}_0} = \frac{1}{\binom{n}{k}^2} \sum_J \int \exp\left(-\sigma_1^2/2 + \sigma_1 <\mathcal{A}_1, \mathcal{X}>\right) \mu(d\mathbf{v}_1) \tag{127}$$

Squaring (127), we have

$$\Lambda^2 = \frac{1}{\binom{n}{k}^4} \sum_{J,J'} \exp(-\sigma_1^2) \int \exp\left(\sigma_1 <\mathcal{A}_1 + \mathcal{A}_1', \mathcal{X}>\right) \mu(d\mathbf{v}_1)\mu(d\mathbf{v}_1') \tag{128}$$

Therefore using Lemma 10 we have

$$\mathbb{E}_0[\Lambda^2] = \frac{1}{\binom{n}{k}^4} \sum_{J,J'} \int \exp\left(\sigma_1^2/2 <\mathcal{A}_1, \mathcal{A}_1'>\right) \mu(d\mathbf{v}_1)\mu(d\mathbf{v}_1') \tag{129}$$

$$= \frac{1}{\binom{n}{k}^2} \sum_J \int \exp\left(\sigma_1^2/2 <\mathcal{A}_1, \mathcal{A}_{fixed}>\right) \mu(d\mathbf{v}_1)$$

where in the last step we used the rotational invariance of probability measures. $\mathcal{A}_{fixed}$ is a fixed tensor with $J_1' = J_2' = [k]$ and $\mathbf{v}_1' = \mathbf{e}_1$. Let $\rho_{J_1}$ be the overlap ratio of $J_1$ with $J_1' = [k]$:

$$\rho_{J_1} \triangleq \frac{|J_1 \cap J_1'|}{k}. \tag{130}$$

$\rho_{J_2}$ is defined similarly. Thus, using Lemma 11 we have

$$\mathbb{E}_0[\Lambda^2] = \frac{1}{\binom{n}{k}^2} \sum_J \int \exp\left(\sigma_1^2 \rho_{J_1}\rho_{J_2} <\mathbf{v}_1, \mathbf{e}_1>/2\right) \mu(d\mathbf{v}_1) \tag{131}$$

$$\leq c \sum_{\rho_{J_1}, \rho_{J_2}} \mathbb{P}(\rho_{J_1} = \rho_1, \rho_{J_2} = \rho_2) \exp(\sigma_1^4 \rho_1^2 \rho_2^2/2m)$$

where $c$ is a constant. Let $a = \rho_1 k$. We have

$$\mathbb{P}(\rho_{J_1} = \rho_1) = \frac{\binom{k}{a}\binom{n-k}{k-a}}{\binom{n}{k}} \leq \frac{\exp\left(a\log(\frac{ek}{a})\right)\exp\left((k-1)\log(\frac{e(n-k)}{k-a})\right)}{\exp\left(k\log(\frac{n}{k})\right)} \tag{132}$$

$$\leq c_1 \exp\left(-k\left(\rho_1\log(\frac{n}{k}) + \rho_1\log(\rho_1) + (1-\rho_1)\log(1-\rho_1)\right)\right)$$

$$\leq c_2 \exp\left(-k\rho_1\log(\frac{n}{k})\right).$$

A similar argument can be written for $\mathbb{P}(\rho_{J_2} = \rho_2)$. Under the condition of Theorem MT-8, using (132) in (131) results in a bounded $\mathbb{E}_0[\Lambda^2]$. Then Lemma 2 of [4] completes the proof.