[Reviews · NeurIPS 2017]

Reviewer 1



The paper solves the tensor biclustering using tensor (un)folding+spectral clustering. This is an interesting paper and addresses a relevant problem mainly seem in computational biology/genomics and in domains having multiple data modalities. a) It would benefit the reader if you can mention in the Introduction section itself that you achieve tensor biclustering via tensor (un)folding followed by spectral clustering. b) It took me time to understand Lines 38-43 and it could be that I have misunderstood your intent. Please clarify: We can perform Tensor bi/tri clustering on an asymmetric tensor, right? Given this, could you please explain Line 40. Again, you mention clustering based on similar trajectories. How did you get ‘similar’ trajectories? In triclustering, you get a resultant cuboid in which all the 3 axes are similar based on some similarity measure. From Eq (3), you create a folded tensor but taking into account ‘all’ trajectories. When does the ‘similar’ trajectories kick in? If dimensions of C_2 and C_1 are n2 x n2 and n1 x n1 respectively then how/where are you capturing the m-dimensional similarity structure? c) Minor comment - Eq 2: T_(j_1,1). What does 1 denote?

Reviewer 2



This paper studies the tensor biclustering problem, which is defined as follows: given an order-3 tensor that is assumed to be of rank 1 with additive Gaussian noise, we want to find index sets along with mode 1 and mode 2 where the factor vectors have non-zero entries. The paper considers several methods and analyzes their recovery performance. The statistical lower bound (i.e. the best achievable performance) is also evaluated. The performance is also evaluated by experiments. The paper is clearly written, self-contained, and easy to follow. Overall, this paper aims to develop a relatively minor problem, tensor biclustering, theoretically in depth, and it seems to be succeeded. For computationally efficient methods (the proposed algorithms), the asymptotic order of variance where the non-zero indices are correctly recovered is identified. In addition, the optimal bound of performance is evaluated, which is helpful as a baseline. Still, I feel there is room for improvement. 1. The model formulation (1) looks less general. The strongest restriction may be the setting q=1. Although the paper says the analysis can be generalized into the case q>1 (Line 83), it is not discussed in the rest of paper. For me, the analysis for q>1 is not trivial. It is very helpful if the paper contains discussion on, for q>1, how we can adapt the analysis, how the theoretical results change, etc. 2. Synthetic experiments can be elaborate. I expected to see the experiments that support the theoretical results (e.g. Theorems 1 and 2). For example, for various n and \sigma_1^2, you can plot monochrome tiles where each tile represents the recovery rate at corresponding n and \sigma_1^2 (e.g. the tile is black if the rate is 0 and is white if the rate is 1), which ideally shows the phase transition that the theorems foreshow.